# Characteristic Changes and Potential Markers of Flavour in Raw Pu-Erh Tea with Different Ageing Cycles Analysed by HPLC, HS-SPME-GC-MS, and OAV

**DOI:** 10.3390/foods14050829

**Published:** 2025-02-27

**Authors:** Jiayi Xu, Xiujuan Deng, Yamin Wu, Miao Zhou, Cen Du, Qiaomei Wang, Yuxin Xia, Junjie He, Wenxia Yuan, Wendou Wu, Hongxu Li, Yankun Wang, Tong Li, Baijuan Wang

**Affiliations:** 1College of Tea Science, Yunnan Agricultural University, Kunming 650201, China; 18088111511@163.com (J.X.); 15808869561@163.com (X.D.); m1678447405@163.com (Y.W.); 18724979985@163.com (M.Z.); wqm19850127@163.com (Q.W.); 15140964046@163.com (Y.X.); junjie19991207@163.com (J.H.); yuanwenxia2023@163.com (W.Y.); 13085315910@163.com (H.L.); wyk18787295901@163.com (Y.W.); 2Tea Distribution Association in Wenshan Prefecture, Wenshan 663000, China; 13577059654@163.com; 3Yunnan Organic Tea Industry Intelligent Engineering Research Center, Yunnan Agricultural University, Kunming 650201, China; wuwd2004@126.com; 4Yunnan Key Laboratory of Crop Production and Smart Agriculture, Kunming 650201, China

**Keywords:** raw Pu-erh tea, ageing cycles, quality, flavour, OAV

## Abstract

To investigate the flavour evolution mechanism of raw Pu-erh tea (RPT) during storage, the volatile and non-volatile compounds of RPT with different storage years (1–10 years) from the same raw material origin, manufacturer, and storage location in Wenshan Prefecture, Yunnan Province, were systematically analysed by HPLC, HS-SPME-GC-MS, and OAV. The results showed that both cluster analyses based on non-volatile and volatile compounds could classify RPT of different storage years into three ageing cycles, with key turning points in the third and eighth years of storage, which is also accompanied by the colour changing from green to orange or brown, the aroma changing from a faint scent to woody and ageing, the astringency diminishing, and the sweet and mellow increasing. Theophylline was identified as the potential marker of RPT stored 1–3 years, while (−)-catechin gallate, (−)-gallocatechin gallate, quercetin, and rutin as those for a storage of 9–10 years. The volatile compounds indicate a general trend of an initial increase followed by a decrease. Forty-four key aroma compounds (OAV ≥ 1) were identified. Eucalyptol, β-Caryophyllene, 2-Amylfuran, Copaene, Estragole, and α-Terpinene originated as potential markers for RPT stored 1–3 years, while (Z)-Linalool oxide (furanoid), α-Terpineol, Terpinen-4-ol, and cis-Anethol were for RPT stored 8–10 years. This study revealed the flavour characteristics and quality changes of RPT over the course of storage, and constructed a sensory flavour wheel, providing theoretical underpinnings for the quality control and assessment of RPT.

## 1. Introduction

Tea drinking in China boasts a history exceeding 1000 years, with Yunnan Province recognised as the origin of the tea plant [1]. Pu-erh tea, recognised with Protected Geographical Indication status, is derived from sun-dried green tea made from the large-leaf varietals of the Yunnan province. According to different processing techniques and quality characteristics, Pu-erh tea is categorised into two varieties: raw Pu-erh tea and ripe Pu-erh tea [2,3]. Pu-erh tea is popular for its weight and fat reduction, sugar reduction, uric acid reduction, anti-fatigue, antioxidant, anti-cancer, and free radical scavenging effects [4,5,6,7]. At the same time, the characteristic of “the more it ages, the more fragrant it is” adds unique enjoyment to the Pu-erh tea and makes it a good collector’s item [8]. Raw Pu-erh tea is produced by steaming and pressing sun-dried green tea from large-leafed varieties in Yunnan Province. Its quality is characterised by a pure and long-lasting aroma, as well as a rich and sweet taste. In scientific warehousing and with ageing time, its taste will evolve towards being mellow, sweet, thick, and smooth. The aroma will transform from a clear fragrance to a woody, sweet, and unique aged aroma [9]. The ageing of raw Pu-erh tea involves the interaction of various factors, including microorganisms, enzymes, humidity, temperature, and oxidation. These factors contribute to a series of chemical changes in the tea under specific environmental conditions, giving rise to the qualities characteristic of raw Pu-erh tea [10]. The quality of Pu-erh tea is intricately linked to both the geographical origin of its raw materials and the conditions of its storage, in addition to the duration of storage. Therefore, it is essential to investigate the effect of storage years on the quality characteristics of raw Pu-erh tea, along with its implications for both drinking and commercial value.

At present, numerous studies have explored the internal mechanisms of raw Pu-erh tea ageing across different storage years and analysed the reasons for the quality differences in raw Pu-erh tea. In terms of sensory quality, Jiao [11] studied the regular changes of quality indicators during the storage process of raw Pu-erh tea. The results showed that, with the increase of storage time, the colour of the soup of raw Pu-erh tea deepened gradually, the aroma changed from fresh to aged, the taste changed from mellow to thick, and the colour of the bottom of the leaves became deeper and darker. The sensory quality improved after 10 years of storage and above but began to decline after 18 years of storage. Shifting to non-volatile compounds, Liu [12] then discovered that as the ageing period extended, free amino acids and tea polyphenols decreased, whereas water-soluble proteins and tea pigments increased in raw Pu-erh tea. Duan [13] reported an upward trend in the water extract of raw Pu-erh tea with lengthening storage durations, while caffeine content exhibited a downward trend. Furthermore, Zhou [14] determined that storing raw Pu-erh tea under hot and humid conditions promoted the degradation of ester catechins and the concurrent accumulation of flavonoids and GA. With respect to volatile compounds, Rong [15] utilised an integrated approach involving GC-E-Nose, GC-MS, and GC-IMS to analyse the change of the volatile components of Pu-erh tea over different storage periods; nine volatile constituents, including Linalool and (E)-2-Hexenal, were selected as key variables for differentiating Pu-erh teas based on storage duration. Ma [16] utilised multivariate statistical methods to analyse the aroma composition of raw Pu-erh tea from Lincang City; a clear distinction could be observed between samples originating from the southern and northern regions of Lincang City. Southern samples were represented by relatively high concentrations of alcohols and esters, whereas northern samples displayed higher levels of aldehydes, ketones, and other substances, the altitude was hypothesised as a potentially significant factor contributing to the observed regional variations in aroma profiles [17]. Xu et al. concluded that the aroma of raw Pu-erh tea is affected by the storage environment. In a dry and cold storage environment, raw Pu-erh tea is more likely to produce floral and sweet aromas. In contrast, in a hot and humid environment, raw Pu-erh tea is more likely to produce woody and aged aromas [18].

Wenshan Prefecture is located in the southeastern part of Yunnan Province, 103°35′~106°12′ E, 22°48′~24°28′ N, with an altitude of 107.02–2991.20 m. It has a subtropical humid monsoon climate. Moreover, Wenshan Prefecture is within the protected area of the Pu-erh Tea Geographical Indication, boasting abundant tea tree planting resources and a wide distribution. It also features significant regional characteristics, such as a rich and fragrant aroma and a fresh and mellow taste [19]. This study focused on raw Pu-erh tea sourced from the same origin, manufacturer, and storage location in Wenshan Prefecture, due to the scarcity of research on tea quality characteristics in the region. It subsequently conducted a thorough and systematic analysis of the impact of storage duration on the quality characteristics of raw Pu-erh tea.

## 2. Materials and Methods

### 2.1. Tea Samples

Samples stored for 1–10 years were all offered by Longli Chun Tea Industry Co., Ltd., located in Wenshan Zhuang and Miao Autonomous Prefecture, Yunnan Province, in China. All samples were produced from Yunnan large-leaf varietals, utilising one bud and two to three leaves as fresh leaf raw material. The production followed the processing method stipulated in GB/T22111-2008 [20], and the products were then uniformly stored in a dry and well-ventilated warehouse located in Wenshan Prefecture, with a relative humidity of ≤70% and an indoor temperature of ≤25 °C. Before further processing, each raw Pu-erh tea sample was ground and sieved through 60-mesh; tea samples were pulverised and kept at −20 °C. Table 1 presents detailed sample information.

### 2.2. Chemicals

The main reagent standards (purity ≥ 98%) employed in this analysis included theophylline, caffeine, gallic acid, (+)-catechin, (−)-catechin gallate, (−)-epicatechin, (−)-epicatechin gallate, (−)-epigallocatechin, (−)-epigallocatechin gallate, (+)-gallocatechin, (−)-gallocatechin gallate, quercetin, rutin, kaempferol, myrcene, and ethyl decanoate, procured from Shanghai Yuanye Biotechnology Co. Anthrone (ACS reagent) (Shanghai, China) and ninhydrin (ACS reagent) were sourced from Sigma-Aldrich Company (St. Louis, MO, USA). Methanol and acetonitrile (all chromatographic grade) were purchased from Thermo Fisher Company (Waltham, MA, USA). All remaining reagents were of analytical grade and obtained from Tianjin Damao Chemical Reagent Factory.

### 2.3. Sensory Evaluation

The sensory evaluation of raw Pu-erh tea samples from different storage years in Wenshan Prefecture was conducted following China’s national standard GB/T23776-2018 “Methodology for Sensory Evaluation of Tea” [21]. Five experienced members (3 females and 2 males, aged between 22 and 55 years old) from the Tea College of Yunnan Agricultural University performed the assessment. Prior to the assessment, the group members received professional training in the evaluation of raw Pu-erh tea in accordance with the “Methodology for Sensory Evaluation of Tea”. For each evaluation, 3 g of tea were placed in a professional evaluation cup, and 150 mL of boiling water was added twice, steeping for 2 min on the first and 5 min on the second. The results were primarily based on the second steeping while also taking into account the first steeping. The members assessed various characteristics of the tea, including appearance (a), aroma (b), liquor colour (c), taste (d), and tea residue (e). Each sample underwent three rounds of evaluation. Ultimately, the group’s assessment data were compiled; subsequently, the quality of the tea samples was quantitatively evaluated using the prescribed formula. The formula is as follows:PRT (total score) = 20% × (a) + 30% × (b) + 10% × (c) +35% × (d) + 5% × (e)

### 2.4. Detection of Non-Volatile Compounds

Tea polyphenol content was determined according to GB/T8313-2018, “Determination of Total Polyphenols and Catechins Content in Tea” [22]. Free amino acid content was determined according to GB/T8314-2013, “Tea–Determination of Free Amino Acids Content” [23]. The content of water extracts was determined in accordance with GB/T 8305-2013, “Tea–Determination of Water Extracts Content” [24]. The content of soluble sugar was determined by the anthrone-sulfuric acid colorimetric method, and the contents of theaflavins, thearubigins, and theabrownins were measured by the system analysis method [25]. The measurements using the above methods were all carried out on a UV-2102PC spectrophotometer (Element Analytical Instrument Co., Ltd., Shanghai, China).

The determination of catechin, flavonoid, and purine alkaloid fractions was carried out by the high-performance liquid chromatography (HPLC) with reference to the method described by Nian [26]. The instrument used in this study is a 1200 high-speed liquid chromatograph equipped with a C18 column (4.6 mm × 100 mm, 2.7 µm, Agilent (Santa Clara, CA, USA)). This method employed a mobile phase A consisting of 0.261% phosphoric acid and 5% acetonitrile, and a mobile phase B of 80% methanol. The elution gradient proceeded as follows: mobile phase B increased linearly from 10% to 45% between 0 and 16 min; from 16 to 22 min, mobile phase B increased linearly from 45% to 65%; mobile phase B was held constant at 65% from 22 to 25.9 min; from 25.9 to 29 min, mobile phase B increased linearly to 100%; and mobile phase B was maintained at 100% from 29 to 30 min. The column temperature was maintained at 35 °C. Each sample was extracted and analysed in triplicate.

### 2.5. Detection of Volatile Compounds

Headspace solid-phase microextraction (HS-SPME) combined with gas chromatography–mass spectrometry (GC-MS) was utilised to separate and identify the volatile compounds of raw Pu-erh tea from Wenshan Prefecture with different storage years. The instruments used in this study include a 7890A-5975C headspace solid-phase microextraction GC-MS (Agilent, USA), a DB-WAX column (30 m × 0.25 mm × 0.25 µm, Agilent, USA), and a 65 µm solid-phase microextraction head (PDMS/DVB, Supelco (Bellefonte, PA, USA)).

Headspace solid-phase microextraction: 1 g of tea sample was placed in a 20 mL headspace vial. A total of 1 µg of ethyl acetate (decanoic acid ethyl ester) was added as the internal standard, followed by the addition of 6 mL of boiling water. The mouth of the vial was then sealed. CTC autosampler: 60 °C, stabilised for 10 min; 65 µm polydimethylsiloxane/divinylbenzene (PDMS/DVB) extraction head; 60 °C extraction for 30 min at a rotational speed of 250 rpm. Gas chromatography–mass spectrometry (GC-MS) coupled detection, with an inlet port temperature of 230 °C and a desorption time of 5 min.

GC conditions were as follows: column: DB-WAX (30 m × 0.25 mm × 0.25 µm); carrier gas: He; oven: 50 °C (5 min) to 230 °C (7 min) at 6 °C/min; column temperature: 50 °C (5 min) to 230 °C (7 min), temperature rising rate: 6 °C/min; split ration: no split; MS conditions: mass spectrometry conditions; ion source: EI; gas interface temperature: 280 °C; and ion source temperature: 230 °C; quadrupole temperature: 150 °C.

Qualitative and quantitative techniques were employed to identify and measure volatile compounds, following the approach outlined by Deng et al. [27,28]. The preliminary detection of volatile compounds was performed utilising the NIST 17 mass spectral library from the National Institute of Standards and Technology. Furthermore, the identity of each compound was confirmed by matching its retention index (RI). The relative mass concentration of each compound was calculated using the internal standard semi-quantitative method. Furthermore, the odour activity value (OAV) is calculated by the ratio of the relative mass concentration of the volatile component to its olfactory threshold in water.

### 2.6. Data Analysis

Data standardisation was performed with Microsoft Excel 2021, with standard deviation presented as (±). Compound content significance was determined (*p* < 0.05), along with a fold change calculation. Principal component analysis (PCA) and orthogonal partial least squares discriminant analysis (OPLS-DA) were conducted using SIMCA 14.1. Data visualisation and representation were achieved with Origin 2021, MetaboAnalyst 6.0, TBtools V2.1, and Chiplot (www.chiplot.online). Online, Excel 2019, and PowerPoint 2019.

## 3. Results and Discussion

### 3.1. The Evolution of Sensory Flavour of RPT During Storage

The sensory flavour wheel of RPT from Wenshan Prefecture across different storage durations is presented in Figure 1. The sensory evaluation scoring outcomes are presented in Table 2. It was noted that, as storage duration extends, the elements such as the colour, aroma, and taste of the RPT have experienced enriching changes. Appearance colour transitions from green to yellow and finally to auburn, with the auburn hue intensifying over time. Soup colour follows a similar progression from green to yellow and then to orange, indicating a gradual increase in chroma. The aroma, initially a faint and sweet scent, develops woody notes and eventually exhibits ageing characteristics. The ageing aroma intensifies with extended storage, with Y1 and Y9 indicating particularly robust and persistent aromatic qualities. Taste evolves from an initial astringency to sweetness, culminating in a mellow character. The astringent flavour gradually dissipates as mellowness becomes more clear. The colour of the tea residue similarly shifts from green to yellow and finally to auburn, its intensity deepening with age.

### 3.2. Dynamic Changes of Non-Volatile Compounds in RPT Stored for Different Years

According to Figure 2A,B, which presents PCA and clustered heat map analyses of 22 non-volatile compounds, three ageing cycles for RPT from the Wenshan Prefecture stored for 1–10 years are indicated: Group A1 (stored for 1–3 years), Group B1 (stored for 4–8 years), and Group C1 (stored for 9–10 years). With the prolongation of storage duration, the contents of 22 non-volatile compounds in the three groups of RPT samples can be categorised into four changing trends. The first trend of change showed a gradual decrease, including the content of TP, ECG, and Theo (Appendix A). The second trend of change showed a fluctuating decrease, including the content of FAA, SS, CA, EGCG, EGC, EC, GC, and Myr (Appendix A). According to previous authors [24,25], changes in the content of amino acids, tea polyphenols, catechins, flavonoids, purine alkaloids, etc., affect the taste characteristics of tea broth, including freshness, bitterness, astringency, and sweetness. Therefore, the reduction in the content of TP, ECG, EGCG, EC, GC, Theo, and other bitter compounds will lead to a decrease in bitterness and astringency, as well as a reduction in the intensity of the taste of RPT from different storage years in Wenshan Prefecture. On the contrary, the third trend showed a gradual increase, including the contents of WE, TF, TR, TB, CG, GCG, Que, and Rut (Appendix A). The fourth trend showed a fluctuating increase, including the contents of Kea, C, and GA (Appendix A). The inherent instability of catechins renders them susceptible to degradation, oxidation, polymerisation, and isomerisation during storage. These reactions, potentially interacting in competitive or synergistic relationships, likely occur simultaneously, causing varying degrees of change in the catechin content of RPT from Wenshan Prefecture across different storage durations [29,30]. Such changes include the isomerisation of EGCG and ECG, resulting in higher CG and GCG levels. Lv [2] explains that TF, TR, and TB, arising from the hydrolysis, oxidation, and polymerisation of catechins, give the tea liquor its yellow, red, and auburn colours, respectively. These pigments, in conjunction with proteins, influence the appearance of the tea leaves. The progressive darkening of the leaves, liquor, and residue during extended storage can be partly attributed to increasing tea pigment content. Simultaneously, elevated TR and TB levels enhance the harmonious, sweet, and mellow flavour of tea, balancing its bitterness and astringency [31,32]. Potential markers (VIP > 1, FC ≥ 1.5, or FC ≤ 0.8) were further analysed by OPLS-DA and fold change calculations for three groups of RPT with different ageing cycles, yielding that different ageing cycles have different potential markers (Figure 2C). Among them, Theo is a potential marker for distinguishing Group B1 and Group C1 from Group A1. FAA, EGC, and GC are potential markers for distinguishing Group B1 from Group A1. CG, GCG, Que, and Rut are potential markers for distinguishing Group A1 and Group B1 from Group C1. TF and TR are potential markers for distinguishing Group A1 from Group C1. TB is a potential marker for distinguishing Group A1 from Group C1, and C is a potential marker for distinguishing Group C1 from Group B1. These evolving characteristics likely contribute to the unique regional flavour and allow for differentiation based on the storage duration of RPT (Appendix A).

### 3.3. The Evolution of Volatile Compounds in RPT Stored for Different Years

#### 3.3.1. The Volatile Profile of RPT

Based on HS-SPME-GC-MS, a total of 184 volatile compounds were detected in RPT from Wenshan Prefecture across a range of storage periods. According to Figure 3A, the total concentration of these compounds exhibited a general trend of increasing and then decreasing with longer storage. More specifically, the total volatile compound concentration rose sharply, reaching its peak in samples stored for 3 years. A sharp decline was observed in samples stored for 4–5 years, followed by a slight rebound in samples stored for 6–8 years, and a further decrease in samples stored for 9–10 years, creating a wave-like pattern of decline. This indicates that the third year of storage represents a critical point in the evolution of volatile compounds in RPT from Wenshan Prefecture. It can be seen that the third year of storage is the key turning point in the change of volatile compounds in raw Pu-erh tea from different years in Wenshan Prefecture. This is consistent with the conclusion that 3–4 years of storage ageing is an important turning point for volatile compounds in RPT, as reported by Guo [33]. In addition, the eighth year of storage was the second key turning point in the changes of volatile compounds in RPT from different years in Wenshan Prefecture. The volatile compounds fell into nine distinct categories, including 44 alkenes, 34 alkanes, 24 esters, 20 alcohols, 22 heterocycles, 14 aldehydes, 14 aromatic hydrocarbons, nine ketones, and three other compounds. Their relative mass concentrations accounted for 18.12%, 12.97%, 14.97%, 15.84%, 4.19%, 12.57%, 8.41%, 10.72%, and 2.21% of the total, respectively (Figure 3B) (Appendix A). Among them, olefins had the highest relative mass concentrations (211.10–1533.84 µg/L), followed by alcohols (233.12–1410.99 µg/L). The remaining six categories of volatile compounds also exhibit varying degrees of distribution, and they show different levels of fluctuation changes with the increase of storage years. Figure 3C illustrates that cluster analysis, based on 188 volatile compounds, allows for the categorisation of RPT from Wenshan Prefecture into three groups according to storage duration: Group A2 (Y1, Y2, Y3, 1–3 years of storage), Group B2 (Y4, Y5, Y6, Y7, 4–7 years of storage), and Group C2 (Y8, Y9, Y10, 8–10 years of storage).

#### 3.3.2. Screening of Aroma-Active Compounds in RPT from Different Storage Years

A total of 44 aroma-active compounds (OAV ≥ 1) were identified based on their odour thresholds and aroma profiles (Figure 4). Of these, 42 aroma-active compounds exhibited OAV ≥ 10, while two aroma-active compounds had 10 ≥ OAV ≥ 1 (Figure 5) (Appendix A). The aroma characteristics of these 44 compounds fell into nine descriptive categories: floral, fruity, woody, sweet, green, herbal, nuts, fatty, and roasted (Figure 4). These characteristics have been previously utilised to characterise the aroma of RPT. Moreover, Linalool (OAV = 106,598.48–3875.95, floral), β-Ionone (OAV = 74,251.90–12,348.77, violet, floral, sweet, fruity, woody), D-Limonene (OAV = 14,880.11–2078.29 fruity, sweet, lemon, orange), (Z)-3,7-Dimethylocta-1,3,6-triene (OAV = 2024.92–129.29, floral, herbal, sweet), γ-Terpinene (OAV = 174.85–8.57, citrus, lemon), Cedrol (OAV = 704.85–125.46, woody), (±)-Dihydroactinidiolide (OAV = 971.25–18.20, fruity, ripe apricot, oily, woody), 2,2,6-Trimethyl-cyclohexanone (OAV = 1834.61–120.21, honey, black pepper), Naphthalene (OAV = 481.36–59.70 camphor, tar) and nine other aroma substances were the key aroma active substances (OAV ≥ 1) common to all groups of different storage years samples, which together formed the basis of the overall aroma styles of the different storage years of RPT in Wenshan Prefecture.

From Figure 5 and Appendix A, it can be seen that some aroma-active compounds appeared only at specific storage stages. 2-Amylfura (OAV = 4201.53–15,656.58, roasted, sweet fruity) was exclusively found in tea samples aged 1–3 years and reached the highest value (OAV = 15,656.58) in tea samples from the storage of 3 years. 2-Amylfura is a notable component of the ‘honey-orchid’ scent, produced via the Maillard reaction during the pressing and drying of tea [34,35]. Cis-Anethol (OAV: 421.88–3719.92, aniseed, herbaceous, sweet aroma) appeared only in tea samples from 2–4 years of storage and reached its highest value in tea samples from the storage of 3 years (OAV = 3719.92). Cis-Anethol is also the main aroma component of RPT after the addition of exogenous amino acid treatment [36]. (Z)-Linalool oxide (furanoid) (OAV = 66.90–538.16, strong woody and floral, camphor) appeared only in tea samples from 5–10 years of storage and reached its highest value in tea samples from the storage of 8 years (OAV = 538.16). According to the previous authors, (Z)-Linalool oxide (furanoid) is not only the distinguishing aroma-active compound of PRT in different storage environments but also the characteristic aroma-active compound of white tea, black tea, rock tea, and other tea types in different storage years [37,38,39,40]. At the same time, there are also some aroma-active compounds that are only present in separate storage years, including 3-Nonen-2-one (OAV = 11.61, fruity, spicy, liquorice odour) in the 1-year storage tea sample, 3-Carene (OAV = 107.62, pine odour) in the 2-year storage tea sample, Estragole (OAV = 4146.70, herbal aroma), Limonene (OAV = 4146.70, fruity, berry), (+)-α-Pinene (OAV = 2462.64, strong woody, piney odour), and Dihydrocarvone (OAV = 3.96, herbal, lingonberry-like odour) in the storage 8-year tea sample. Moreover, 28 aroma-active compounds reached their maximum values at the first critical turning point, i.e., tea samples from 3 years of storage, including Linalool (OAV = 106,598.48), β-Cyclocitral (OAV = 87,428.53), and β-Ionone (OAV = 74,251.90) with OAV ≥ 10,000, Safranal (OAV = 73,552.80), Copaene (OAV = 28,492.76), Eucalyptol (OAV = 23,073.88), 1-Octen-3-ol (OAV = 16,220.49), 2-Amylfuran (OAV = 15,656.58), D-Limonene (OAV = 14,880.11), Nonanal (OAV = 10,023.89), and other aroma-active compounds with floral, sweet, fresh, fruity, and roasted aroma characteristics. However, eight aroma-active compounds reached their maximum values at the second critical turning point, i.e., tea samples from the storage year 8, and they were Limonene with OAV ≥ 1 (OAV = 3383.37), (+)-α-Pinene (OAV = 2462.64), (E)-Linalool oxide (furanoid) (OAV = 541.16), (Z)-Linalool oxide (furanoid) (OAV = 538.16), α-Terpineol (OAV = 319.06), Terpinolene (OAV = 319.06), Terpinen-4-ol (OAV = 24.89) Dihydrocarvone (OAV = 3.96) and other aroma-active compounds with fruity, woody and herbal aroma characteristics.

#### 3.3.3. Changes of Aroma-Active Compounds in RPT with Different Storage Ageing Cycles

With increasing storage duration, the aroma-active compounds and their corresponding OAV values shift across different ageing stages, resulting in differences in the aroma type characteristics of RPT from Wenshan Prefecture in each storage group. In group A2 (1–3 years of storage), 33 aroma-active compounds exhibited higher levels compared to group B2 and C2. Among the most potent aroma-active compounds, β-Ionone (OAV = 52280.17, violet, floral, sweet, fruity, woody) presented the highest OAV, succeeded by β-Cyclocitral (OAV = 52,259.92, sweet, fruity, herbal), Linalool (OAV = 41,921.32, floral), Safranal (OAV = 24,517.6, saffron, herbal, woody), Copaene (OAV = 20,053.18, honey, spicy, woody), Eucalyptol (OAV = 12,582.03, mint), 2-Amylfuran (OAV = 9981.23, roasted aroma, sweet fruity), D-Limonene (OAV = 7291.68, sweet fruity, lemon, orange), Nonanal (OAV = 5678.43, light and sweet rose aroma, citrus), and 1-Octen-3-ol (OAV = 5406.83, greasy, mushroom, floral) (Figure 6) (Appendix A). These compounds, contributing floral, green, sweet, fruity, herbal, and roasted aromatic notes, help form the aroma of ’faint scent and sweet’ in the A2 tea samples.

In group B2 (4–7 years of storage), Linalool (OAV = 26,077.19, floral) presented the highest odour activity value, succeeded by β-Ionone (OAV = 16019.8, violet, floral, sweet, fruity, woody) and β-Cyclocitral (OAV = 52,259.92, sweet, fruity, herbaceous). These findings align with the top aroma-active compounds observed in A2. Moreover, aroma-active compounds arose from B2 compared to A2: Terpinolene (OAV = 135.85, herbal aroma, woody), 1-Methyl-Naphthalene (OAV = 815.36, camphor, herbal aroma, chemical), Terpinen-4-ol (OAV = 7.57, woody), o-Cymene (OAV = 1.58 floral), (Z)-Linalool oxide (furanoid) (OAV = 58.57, strong woody and floral aroma, camphor), and (E)-Linalool oxide (furanoid) (OAV = 52.97 woody, floral); whereas, Naphthalene (OAV = 100.78, camphor, tar), β-Ionone (OAV = 16,019.8, violet, floral, sweet, fruity, woody), β-Cyclocitral (OAV = 52,259.92 sweet, fruity, herbaceous), and Nonanal (OAV = 563.68, fresh sweet roses, citrus) demonstrated lower OAVs in later B2 relative to A2 (*p* < 0.05). In addition, 2-Methyl-Naphthalene (OAV = 2729.03, sweet, floral, woody) and cis-Anethol (OAV = 578.85, herbal, aniseed, sweet) displayed higher OAVs in B2 compared to the tea samples from group C2 (Figure 6) (Appendix A). Overall, relative to the A2 group, the B2 group demonstrated a decrease in aroma-active compounds associated with sweet, fruity, floral, and green aromas, with an increase in compounds contributing to herbal and woody aromas. This shift resulted in the development of a ’woody with sweet’ aroma profile characteristic of the B2 tea samples. The woody aroma is characteristic of Pu-erh tea and serves as a significant aromatic attribute of Pu-erh tea stored in hot and humid circumstances [18].In group C2 (8–10 years of storage), Linalool (OAV = 29,107.94, floral) presented the highest odour activity value among aroma-active compounds, followed by β-Ionone (OAV = 29,107.94, violet, floral, sweet, fruity, woody) and Safranal (OAV = 18,132.77, saffron, herbal, woody). Furthermore, several other aroma-active compounds attained notable OAVs: Terpinolene (OAV = 140.31, herbal aroma, woody), 1-Methyl-Naphthalene (OAV = 1227.6, camphor, herbal aroma, chemical), Terpinen-4-ol (OAV = 16.8, woody), o-Cymene (OAV = 1.96, flowery), (Z)-Linalool oxide (furanoid) (OAV = 323.2, strong woody and floral odour, camphor), (E)-Linalool oxide (furanoid) (OAV = 306.41, woody, floral odour), (+)-α-Pinene (OAV = 820.88, pine, coniferous, resinous, strong woody and piney odour), Limonene (OAV = 1127.79, fruity, berries), Dihydrocarvone (OAV = 1.32, herbaceous, such as spearmint aroma), α-Terpineol (OAV = 203.73, orchid, light, fruity, banana), and Hexanal (OAV = 2485.41, herbaceous, fruity). These compounds reached their peak OAVs in the three tea sample groups. Terpinen-4-ol (OAV = 16.8, woody) exhibited a significantly greater OAV compared to group B2, while (Z)-Linalool oxide (furanoid) (OAV = 323.2, woody, floral) exhibited a significantly greater OAV compared to both A2 and B2 (*p* < 0.05) (Figure 6) (Appendix A). The increased presence of woody aroma-active compounds, coupled with the diverse array of other aroma-active compounds, contributed to the characteristic ’ageing’ aroma profile of the C2 tea sample. Moreover, no methoxy aroma-active compounds, commonly associated with ’Ageing aroma’ characteristics in Pu-erh tea from previous studies, were detected in C2. However, Shen [37] stated that the Maillard reaction, or the transformation of certain aroma-active compounds during storage ageing, leads to the formation and accumulation of the aroma of herbals, wood, and medicinal herbs. Therefore, both the foundational aroma-active compounds of RPT and those from the later transformation process originate the ’Ageing aroma’ characteristic of RPT.

#### 3.3.4. Potential Markers Responsible for Aroma Differences of RPT in Three Ageing Cycles

From the above, it can be seen that the types and contents of aroma active compounds varied greatly among RPT from Wenshan Prefecture at different cycles of storage and ageing times. Based on OPLS-DA analysis combined with fold change for volatile compounds, firstly, a total of 67 significantly different volatile compounds (VIP > 1, FC ≥ 1.5, or FC ≤ 0.8) were screened (Figure 7) (Appendix A). In Group B2, relative to Group A2, 40 volatile compounds exhibited down-regulation, while 2 demonstrated up-regulation. Similarly, compared to group A2, 40 volatile compounds were down-regulated and 2 volatile compounds were up-regulated in group B2. In Group C2, compared to Group A2, 37 volatile compounds were down-regulated and 6 were up-regulated. In addition, contrasting Group C2 with Group B2 indicated down-regulation of 6 volatile compounds and up-regulation of 29. Specifically, 12 volatile compounds demonstrated significant differences across all three groups. The relative mass concentrations of both Azulene and Pentanoic acid, 2,2,4-trimethyl-3-carboxyisopropyl, isobutyl ester, decreased significantly with extended storage; whereas, Nonanal, β-Ionone, β-Cyclocitral, Diisobutyl phthalate, Dibutyl phthalate, Phytone, 1-Methylene-1H-indene, 2,2,6-Trimethyl-cyclohexanone, 2,4,4-trimethylpentane-1,3-diyl bis(2-methylpropanoate), and 1-Chloro-heptacosane exhibited an initial decrease followed by an increase in relative mass concentration, reaching their lowest levels in Group B2 compared to Groups A2 and C2. As shown in Figure 8, Eucalyptol, β-Caryophyllene, 2-Amylfuran, Copaene, Estragole, and α-Terpinene were the potential aroma markers for group A2 to differentiate between group B2 and group C2. (Z)-Linalool oxide (furanoid), α-Terpineol, and Terpinen-4-ol were potential aroma markers for group C2 to differentiate group A2 from group B2. The aroma potential markers for group A2 to differentiate group B2 were Naphthalene, (Z)-3,7-Dimethylocta-1,3,6-triene, and Cedrol. The aroma potential markers for group A2 to differentiate group C2 were Linalool and 2-Carene. The aroma potential markers for group C2 to distinguish group B2 were (E)-Linalool oxide (furanoid) and Limonene. Additionally, β-Ionone, β-Cyclocitral, Nonanal, 2,2,6-Trimethyl-Cyclocitral, 2,2,6-Trimethyl-Cyclohexanone, and Dibutyl phthalate were the characteristic difference markers shared by all three groups. Moreover, β-Ionone, Linalool, β-Cyclocitral, (E)-Linalool oxide (furanoid), and Limonene have been shown to be potential markers of aroma in Pu-erh tea [27,41], and 2,2,6-Trimethyl-cyclohexanone is an intermediate that can be used to synthesise β-Ionone [42]. In previous studies, the variety of tea, the technology of processing, and the conditions of storage, including pathways such as oxidation and degradation of compounds caused by the environment, affect the formation of potential markers of aroma [43,44].

#### 3.3.5. Metabolic Evolutionary Pathways of Major Aroma-Active Compounds

The above studies have shown that the aroma characteristics of RPT change during storage due to the influence of various factors. Generally, glycosides, fatty acids, amino acids, carbohydrates, and carotenoids are significant aroma precursors for the synthesis of volatile chemicals in tea [45]. Leveraging KEGG data, key aroma compound metabolic routes were investigated further. Through the results of the study, it was found that the aroma-active compounds of RPT from Wenshan Prefecture were mainly synthesised as monoterpenes, which are usually woody, floral and fruity, and favorably impacts RPT’s scent profile [46], including Linalool, β-Ionone, β-Cyclocitral, and Safranal. Previous research indicates that the synthesis of these compounds in tea occurs primarily via the metabolic transformation of two principal precursors: isopentenyl diphosphate and di-methylallyl pyrophosphate. Furthermore, the 2-C-methyl-D-erythritol-4-phosphate pathway and the carotenoid cleavage enzyme play critical roles in the production of monoterpenes and the oxidation of carotenoids [47]. Among them, Linalool oxides will be produced due to Linalool through an oxidoreductase reaction, including (Z)-Linalool oxide (furanoid) and (E)-Linalool oxide (furanoid) [48], and the production of these Linalool oxides helps to promote the wood flavour of RPT. β-primeverosides and β-glucopyranosides are the main precursors for the biosynthesis of Linalool and Linalool oxides in tea, which are produced by degradation and hydrolysis of the corresponding enzymes [48]. β-Ionone, terpene aldehyde, and terpene ketone are carotenoids that are produced by enzymatic and thermo-chemical reactions, and β-Cyclocitral is also an important intermediate in the synthesis of Safranal [44].

## 4. Conclusions

This study utilised sensory evaluation, HPLC, and HS-SPME-GC-MS, coupled with multivariate statistical analysis, to elucidate the evolving quality and characteristic profiles of RPT from different storage years in Wenshan Prefecture. This combined analytical approach enabled the classification of samples into three groups reflecting different ageing cycles based on storage duration. In this way, it illustrates that during the storage process, the sensory characteristics, non-volatile components, and aroma-active compounds of RPT will undergo differential changes over time.

Among them, changes in catechins, theophyllines, flavonoids, purine alkaloids, and free amino acids were potential markers for distinguishing the three groups of raw Pu-erh tea based on their ageing cycle. In addition, CG, GCG, Que, and Rut were potential markers in group C1 as storage time increased. A total of 184 volatile chemicals were identified using HS-SPME-GC-MS, with their overall quantity exhibiting an initial increase followed by a subsequent decrease across tea samples of varying storage years. Combined with OAV analyses of Linalool, β-Ionone, D-Limonene, (Z)-3,7-Dimethylocta-1,3,6-triene, γ-Terpinene, Cedrol, (±)-Dihydroactinidiolide, 2,2,6-Trimethyl-cyclohexanone, and Naphthalene, a total of nine key aroma-active compounds (OAV ≥ 10) together constitute the overall aroma profiles of RPT from different storage years in Wenshan Prefecture. These analyses also identified three sets of aroma-differentiating active compounds associated with different ageing cycles and highlighted (Z)-Linalool oxide (furanoid), α-Terpineol, Terpinen-4-ol, and cis-Anethol as potential markers for group C2. Moreover, β-Ionone, β-Cyclocitral, Nonanal, 2,2,6-Trimethyl-cyclohexanone, and Dibutyl phthalate were identified as common potential markers across the three groups of PRT with varying storage durations from Wenshan Prefecture. This research offers a certain theoretical basis and data support for an in-depth understanding of the quality characteristics of RPT across different storage years and the evolutionary paths of those characteristics. Furthermore, RPT quality is affected by a variety of factors, including harvest season, raw material grade, processing methods, and storage conditions; future studies with larger sample sizes and enhanced variable control are warranted. Integrating metabolomics and microbiological analysis could enhance the investigation of the transformation pathways and processes of compounds during storage ageing. At the same time, advanced technologies such as GC-IMS and GC-O were applied to conduct aroma reconstitution and deletion experiments. These experiments further clarify the characteristic markers, thereby providing more objective bases for the quality assessment of RPT, its flavour characteristics, and the improvement of processing and storage technologies.

## Figures and Tables

**Figure 1 foods-14-00829-f001:**
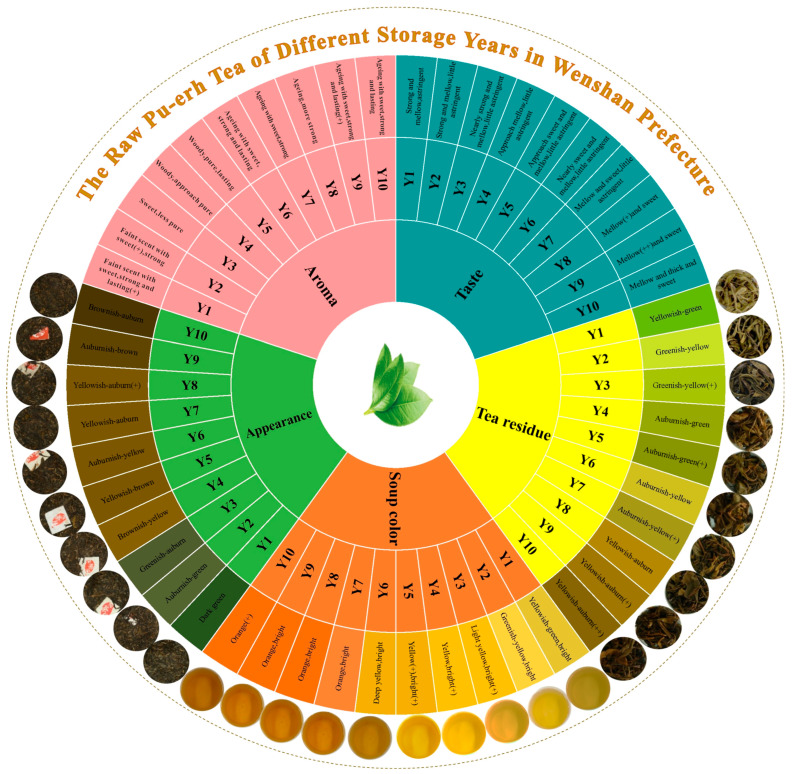
The flavour wheel of raw Pu-erh tea from different storage years in Wenshan Prefecture, Yunnan Province. (+), (++) indicate the degree of the presence of this characteristic.

**Figure 2 foods-14-00829-f002:**
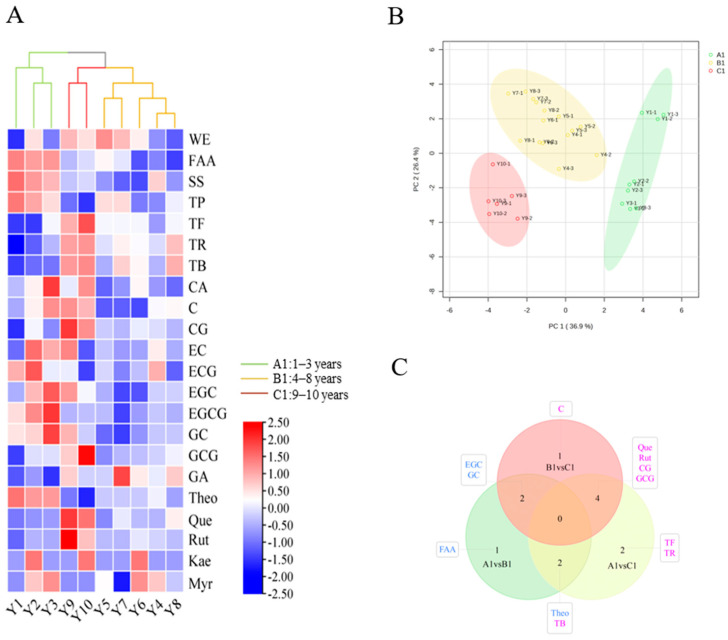
WE: water extracts, FAA: free amino acids, SS: soluble sugars, TP: tea polyphenol, TF: theaflavins, TR: thearubigins, TB: theabrownins, CA: caffeine, C: (+)-catechin, CG: (−)-catechin gallate, EC: (−)-epicatechin, ECG: (−)-epicatechin gallate, EGC: (−)-epigallocatechin, EGCG: (−)-epigallocatechin gallate, GC: (+)-gallocatechin, GCG: (−)-gallocatechin gallate, GA: gallic acid, Theo: theophylline, Que: quercetin, Rut: rutin, Kae: kaempferol, Myr: myrcene. (**A**) Heat diagram of the non-volatile compounds. (**B**) Scatter plot for the PCA model (total). (**C**) Wayne diagrams of potential makers, filter conditions are VIP > 1, FC > 1.5 or <0.8. Blue-labelled compounds were highest in group A1 and purple labelled compounds were highest in group C1.

**Figure 3 foods-14-00829-f003:**
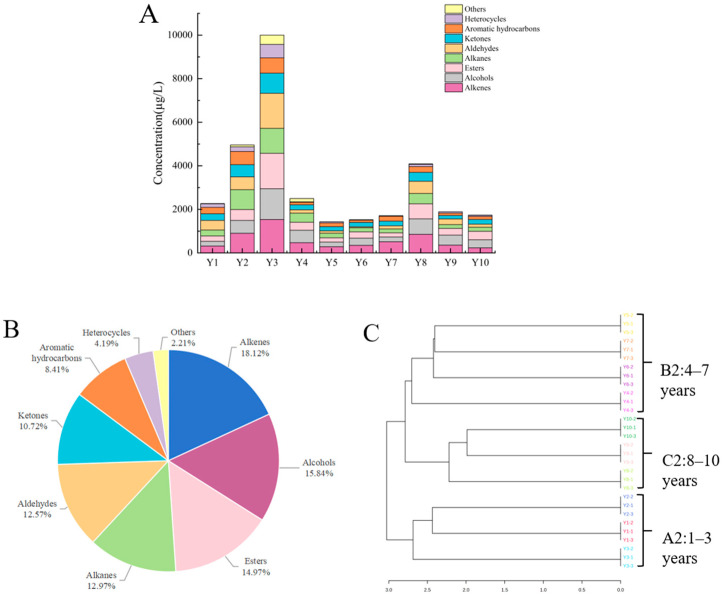
(**A**) The volatile compounds’ concentration histogram in different storage years. (**B**) Percentage of content of volatile compounds by types. (**C**) Cluster dendrogram of VOCs in RPT with different storage years in Wenshan Prefecture.

**Figure 4 foods-14-00829-f004:**
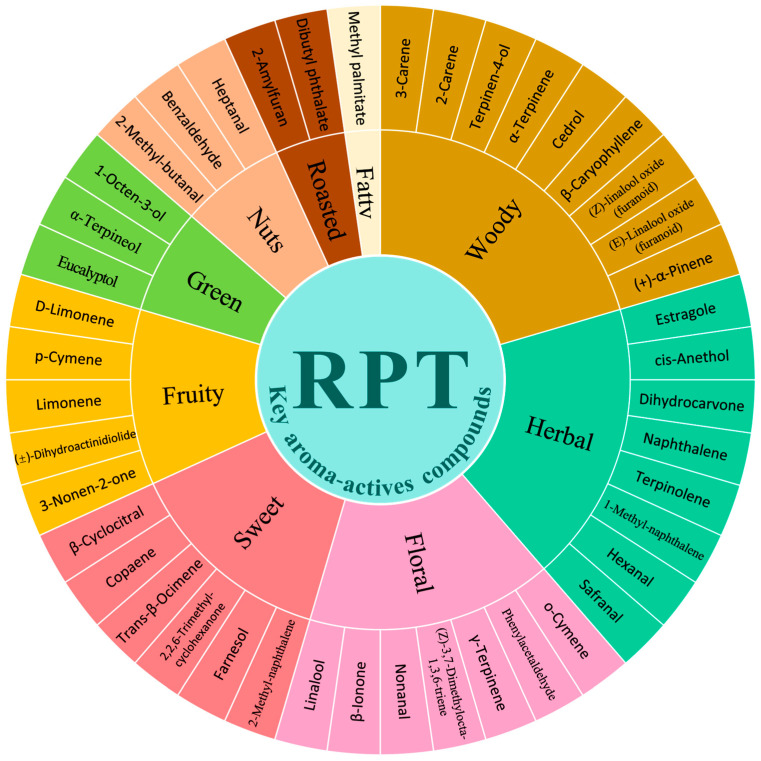
The wheel of key aroma-actives compounds in RPT from different storage years (OAV ≥ 1).

**Figure 5 foods-14-00829-f005:**
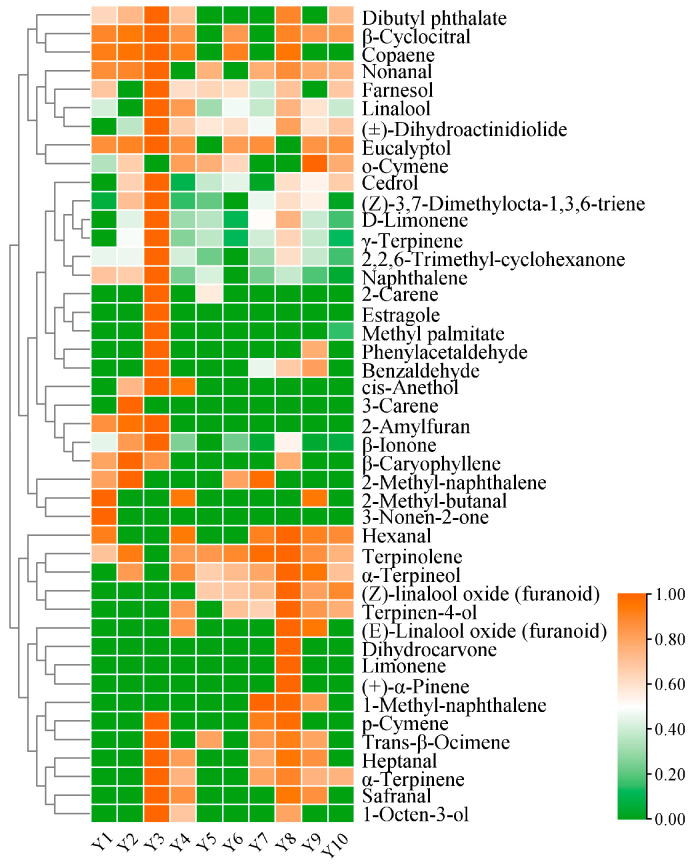
OAV heat map of key aroma-active compounds in PRT of different storage years (OAV ≥ 1).

**Figure 6 foods-14-00829-f006:**
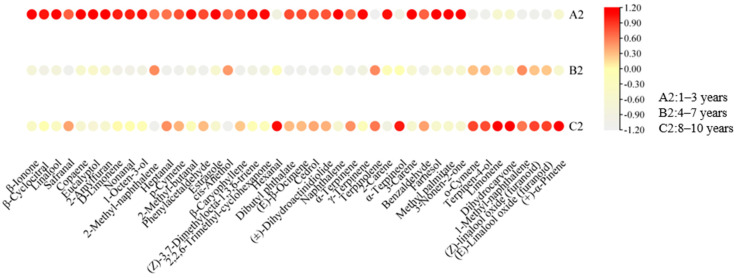
Clustering heat map of key aroma compounds from different storage ageing cycles (OAV ≥ 1).

**Figure 7 foods-14-00829-f007:**
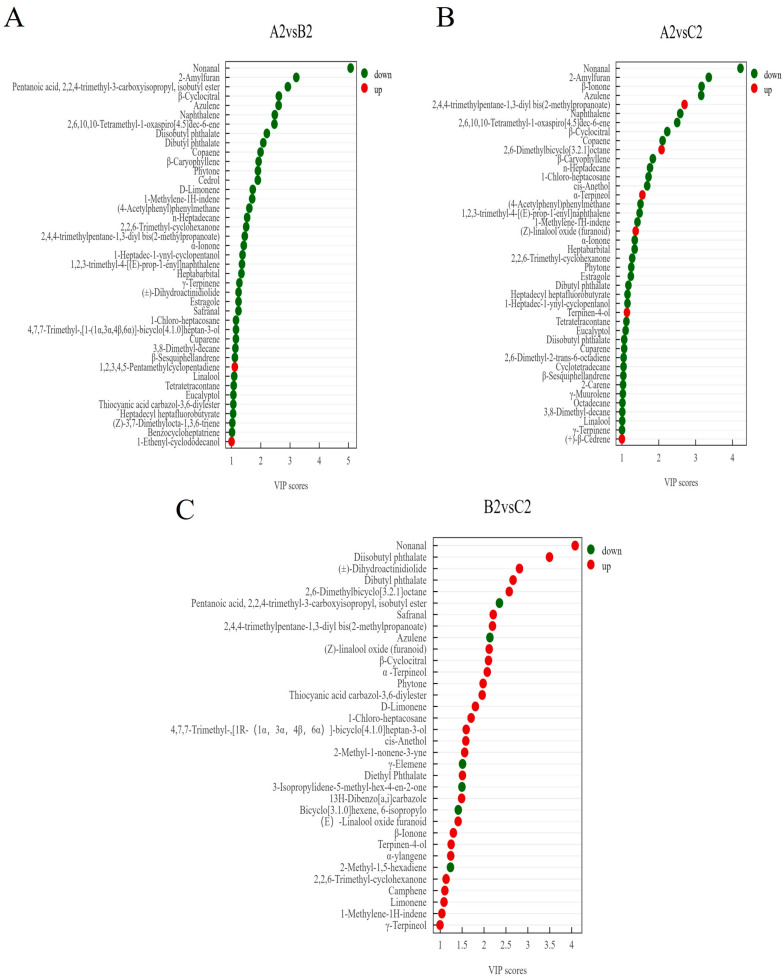
Key differential volatile compounds of RPT in three ageing cycles; filter conditions are VIP > 1, FC > 1.5, or <0.8. (**A**) A2vsB2, green-labelled compounds down-regulated by B2 and red-labelled compounds up-regulated by B2; (**B**) A2vsC2, green-labelled compounds down-regulated by C2 and red-labelled compounds up-regulated by C2; (**C**) B2vsC2, green-labelled compounds down-regulated by C2 and red-labelled compounds up-regulated by C2.

**Figure 8 foods-14-00829-f008:**
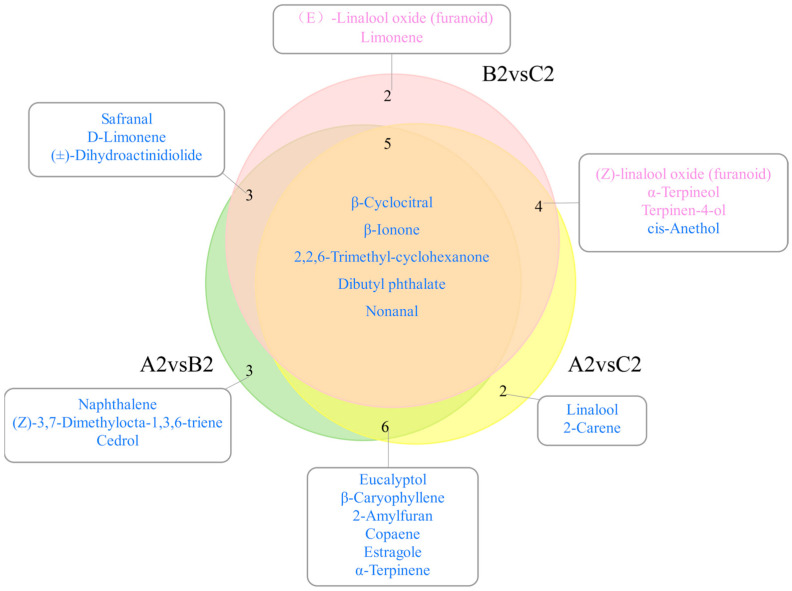
Wayne diagrams of potential differential aroma markers of RPT in three ageing cycles; screening conditions were VIP > 1, FC > 1.5, or <0.8. Blue-labelled compounds were highest in group A2 and purple-labelled compounds were highest in group C2.

**Table 1 foods-14-00829-t001:** Tea samples of raw Pu-erh tea different storage years in Wenshan Prefecture.

No.	Storage Time	Year of Production	Place of Origin	Manufacturer	Storage Location
Y1	1 year	2023	Pengzhai Village, Wenshan Prefecture	Longli Chun Tea Industry Co.	Wenshan Zhuang and Miao Autonomous Prefecture
Y2	2 years	2022			
Y3	3 years	2021			
Y4	4 years	2020			
Y5	5 years	2019			
Y6	6 years	2018			
Y7	7 years	2017			
Y8	8 years	2016			
Y9	9 years	2015			
Y10	10 years	2014			

**Table 2 foods-14-00829-t002:** The sensory evaluation scoring outcomes of raw Pu-erh tea with different storage years in Wenshan Prefecture, Yunnan Province.

Sample	Appearance	s	Aroma	s	Soup Colour	s	Taste	s	Tea Residue	s	Total
Y1	Dark green	94	Faint scent with sweet, strong and lasting ^(+)^	96	Yellowish-green, bright	92	Strong and mellow, astringent	88	Yellowish-green	92	92.20
Y2	Auburnish-green	94	Faint scent with sweet ^(+)^, strong	92	greenish yellow, bright	92	Strong and mellow, little astringent	90	Greenish-yellow	92	91.70
Y3	Greenish-auburn	94	Sweet, less pure	78	Light yellow, bright ^(+)^	94	Nearly strong and mellow, little astringent	86	Greenish-yellow ^(+)^	94	86.40
Y4	Brownish-yellow	92	Woody, approach pure	80	Yellow, bright ^(+)^	94	Approach mellow, little astringent	78	Auburnish-green	92	83.70
Y5	Yellowish-brown	92	Woody, pure, lasting	92	Yellow ^(+)^, bright ^(+)^	94	Approach sweet and mellow, little astringent	78	Auburnish-green ^(+)^	94	87.40
Y6	Auburnish-yellow	92	Ageing with sweet, strong and lasting	94	Deep yellow, bright	92	Nearly sweet and mellow, little astringent	86	Auburnish-yellow	92	90.50
Y7	Yellowish-auburn	92	Ageing with sweet, strong	92	Orange, bright	92	Mellow and sweet, little astringent	90	Auburnish-yellow ^(+)^	94	91.40
Y8	Yellowish-auburn ^(+)^	94	Ageing, more strong	90	Orange, bright	92	Mellow ^(+)^ and sweet	94	Yellowish-auburn	92	92.50
Y9	Auburnish-brown	92	Ageing with sweet, strong and lasting ^(+)^	96	Orange, bright	92	Mellow ^(++)^ and sweet	95	Yellowish-auburn ^(+)^	94	94.35
Y10	Brownish-auburn	92	Ageing with sweet, strong and lasting	94	Orange ^(+)^	90	Mellow and thick and sweet	92	Yellowish-auburn ^(++)^	95	92.55

Notes. (+), (++) indicate the degree of the presence of this characteristic.

## Data Availability

The original contributions presented in the study are included in the article, further inquiries can be directed to the corresponding authors.

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
