# Peer review of "Characteristic Changes and Potential Markers of Flavour in Raw Pu-Erh Tea with Different Ageing Cycles Analysed by HPLC, HS-SPME-GC-MS, and OAV"

_foods, 2025, doi:10.3390/foods14050829_

Round 1

Reviewer 1 Report

Comments and Suggestions for Authors

Line 47, there are double spaces before and after the text with “”.

I feel that the selection of samples with different aging ages is not entirely correct, because being the same plant, same cultivation area, same company, the climatic effects on the crops in different years cannot be controlled. More heat, more cold, more or less environmental humidity, same conditions or concentration of nutrients in the soil, pests, rain, droughts… all these are factors that alter the metabolism of the plant and consequently the accumulation of secondary metabolites, sugars, proteins among other elements of the leaf, and of course it can alter the final product, as well as its flavor, concentration of volatile metabolites and many others. It would have been interesting to analyze, for example, 3 drinks of 10 years, from different years, 3 drinks of 9 years from different years and so on. Something similar to wines, not every year the wines come out the same, despite being from the same vineyards, processes and applied technologies… there are always years that have better quality than others due to the effect of the climate.

Line 128 “2.4. Sensory Evaluation” the letter “n” is missing in evaluation

It is not entirely clear whether aged tea is the plant alone, like the dried and aged plant, or whether it is aged prepared tea. Based on the sensory methodology, I think it is the aged plant alone. Please make it a little clearer.

The image in figure 1 does not allow the text of the results to be read, and I expected the results to be quantitative (a scale of 1 to 5, for example) in order to differentiate the results, in terms of aroma, flavour, appearance...

From my perspective as a reader, I feel that the nomenclature of the bioactive compounds described in point 2.2 does not help in reading the results because there are many nomenclatures, some very similar and it is difficult to compare the variations using the nomenclature, it would be ideal to have the names, for example, in section 3.2.

Line 243 is missing a space at the beginning of the sentence.

Why do they put the names of the compounds to be analysed in figure 3 and not in figure 2? The graphs in figure 3 look very blurry, perhaps they should be included as supplementary information.

I think there is a nomenclature error, in some parts A1, A2, A3 are mentioned and in others A1, B1 and C1. Improve quality of figure 5.

Line 373 is missing space in the text. Review the entire document as this error occurs very frequently.

Author Response

Dear Reviewer.

We feel great thanks for your professional review work on our article. As you are concerned, there are several problems that need to be addressed. According to your nice suggestions, we have made extensive corrections to our previous draft, the detailed corrections are listed below.

Comments 1: Line 47, there are double spaces before and after the text with “”.

Reponse 1: In line 47, the double space before and after “” has been deleted. The corrected sentence reads (line 48): At the same time, the characteristic of “the more it ages, the more fragrant it is” adds unique enjoyment to the Pu-erh tea and makes it a good collector's item [8].

Comments 2: I feel that the selection of samples with different aging ages is not entirely correct, because being the same plant, same cultivation area, same company, the climatic effects on the crops in different years cannot be controlled. More heat, more cold, more or less environmental humidity, same conditions or concentration of nutrients in the soil, pests, rain, droughts… all these are factors that alter the metabolism of the plant and consequently the accumulation of secondary metabolites, sugars, proteins among other elements of the leaf, and of course it can alter the final product, as well as its flavor, concentration of volatile metabolites and many others. It would have been interesting to analyze, for example, 3 drinks of 10 years, from different years, 3 drinks of 9 years from different years and so on. Something similar to wines, not every year the wines come out the same, despite being from the same vineyards, processes and applied technologies… there are always years that have better quality than others due to the effect of the climate.

Reponse 2: Dear Reviewer, thank you very much for your valuable suggestions on our study. Your insights regarding the impact of climatic conditions in different years on tea quality are very perceptive. We also agree with these insights. Indeed, climatic conditions (such as temperature, humidity, and rainfall) can significantly affect the metabolism of plants, thereby influencing the chemical composition and sensory quality of tea.

In our study, we chose samples from different years but from the same origin and company to minimize the variables caused by differences in cultivation conditions. This allows us to more accurately assess the effects of storage time on the flavour and quality of raw Pu-erh tea. Although we acknowledge that different climatic conditions in different years can have some impact on the initial quality of raw Pu-erh tea, the focus of our study is to explore the chemical and sensory changes that occur in tea during long-term storage under the same storage conditions. Regarding your suggestion of “comparing samples from different years,” we understand your point of view. In fact, we have also considered this in our study. The samples we selected all come from the standardized production batches of the same company over the years. Although it is not possible to completely eliminate the differences between years, this design helps us better understand the long-term impact of storage time on tea quality. In addition, we have reviewed a number of previous studies [1-6]. Excluding the influence of climatic conditions on tea metabolism, there are still certain patterns of metabolic changes in tea samples from different years. Therefore, we believe that the sample selection in this study is feasible.

We are very grateful for your comments. We have discussed the potential impact of climatic conditions on the initial quality of tea. Moreover, we believe that these suggestions will be of great reference value for our future experimental design and data analysis. In subsequent studies, we will consider using a more diverse range of samples, such as analyzing multiple batches of samples from different years, to more comprehensively assess the impact of climatic factors on tea quality.

We sincerely thank you for your valuable comments.

References:

  1. Guo, J.; Yu, Z.; Liu, M.; Guan, M.; Shi, A.; Hu, Y.; Li, S.; Yi, L.; Ren, D. Analysis of volatile profile and aromatic characteristics of raw Pu-erh tea during storage based on GC-MS and odour activity value.Foods 2023, 12, 3568.
  2. Zhou, B.; Ma, C.; Wu, T.; Xu, C.; Wang, J.; Xia, T. Classification of raw Pu-erh teas with different storage time based on characteristic compounds and effect of storage environment. Lwt2020, 133, 109914.
  3. Ma B.;Ma C.; Zhou B.; Chen X.; Wang Y.; Li Y.; Yin J.; Li X.; Quantitative descriptive analysis, non-targeted metabolomics and molecular docking reveal the dynamic aging and taste formation mechanism in raw Pu-erh tea during the storage. Food Chem. X, 2025,
  4. Characterization of volatile metabolites in Pu-erh teas with different storage years by combining GC-E-Nose, GC–MS, and GC-IMS.
  5. Chen Z.;Dai W.; **ong M.; Gao J.; Zhou H.; Chen D.; Li Y.; Metabolomics investigation of the chemical variations in white teas with different producing areas and storage durations. Food Chem. X, 2024, 21,
  6. Liang Y.; Liu Z.; Pang Y.; LiM.; Zheng S.; Pan F.; Guo C.; Wu Q.; Chen T.; Li Q.; Liu Z.; Effects of storage durations on flavour and bacterial communities in Liupao tea. Food Chem, 2025, 470,   

Comments 3: Line 128 “2.4. Sensory Evaluation” the letter “n” is missing in evaluation.

Reponse 3: Dear reviewer, a missing ‘n’ has been added to the title. The title has been corrected to reads ( line 123): 2.4. Sensory Evaluation.

Comments 4: It is not entirely clear whether aged tea is the plant alone, like the dried and aged plant, or whether it is aged prepared tea. Based on the sensory methodology, I think it is the aged plant alone. Please make it a little clearer.

Reponse 4: Dear reviewer, Here,the term "aged tea" refers to the storage duration of the tea samples, not to the plant alone. We apologize for the unclear explanation and thank you for pointing this out.

Comments 5: The image in figure 1 does not allow the text of the results to be read, and I expected the results to be quantitative (a scale of 1 to 5, for example) in order to differentiate the results, in terms of aroma, flavour, appearance...

Reponse 5: Dear reviewer, thank you for your suggestion. We have added the quantitative scores of the sensory evaluation results to Table 2 in the manuscript (lines 214-216):

Table 2. The sensory evaluation scoring outcomes of raw Pu-erh tea with different storage years in Wenshan Prefecture, Yunnan Province.

sample

appearance

s

aroma

s

soup color

s

taste

s

Tea residue

s

total

Y1

 Dark green

94

Faint scent with sweet, strong and lasting (+)

96

Yellowish-green, bright

92

Strong and mellow, astringent

88

Yellowish-green

92

92.20

Y2

Auburnish-green

94

Faint scent with sweet (+), strong

92

greenish yellow,

bright

92

Strong and mellow, little astringent

90

Greenish-yellow

92

91.70

Y3

Greenish-auburn

94

Sweet, less pure

78

Light yellow, bright (+)

94

Nearly strong and mellow, little astringent

86

Greenish-yellow (+)

94

86.40

Y4

Brownish-yellow

92

Woody, approach pure

80

Yellow, bright (+)

94

Approach mellow, little astringent

78

Auburnish-green

92

83.70

Y5

Yellowish-brown

92

Woody, pure,lasting

92

Yellow (+), bright (+)

94

Approach sweet and mellow, little astringent

78

Auburnish-green (+)

94

87.40

Y6

Auburnish-yellow

92

Ageing with sweet, strong and lasting

94

Deep yellow, bright

92

Nearly sweet and mellow, little astringent

86

Auburnish-yellow

92

90.50

Y7

Yellowish-auburn

92

Ageing with sweet, strong

92

Orange, bright

92

Mellow and sweet, little astringent

90

Auburnish-yellow (+)

94

91.40

Y8

Yellowish-auburn (+)

94

Ageing, more strong

90

Orange, bright

92

Mellow (+) and sweet

94

Yellowish-auburn

92

92.50

Y9

Auburnish-brown

92

Ageing with sweet, strong and lasting (+)

96

Orange, bright

92

Mellow (++) and sweet

95

Yellowish-auburn(+)

94

94.35

Y10

Brownish-auburn

92

Ageing with sweet, strong and lasting

94

Orange (+)

90

Mellow and thick and sweet

92

Yellowish-auburn (++)

95

92.55

Comments 6: From my perspective as a reader, I feel that the nomenclature of the bioactive compounds described in point 2.2 does not help in reading the results because there are many nomenclatures, some very similar and it is difficult to compare the variations using the nomenclature, it would be ideal to have the names, for example, in section 3.2.

Reponse 6: Thank you for your valuable comments. To improve the readability of the results, we have removed the nomenclature in Section 2.2 and adopted a more comprehensive description of the compounds. Additionally, we have annotated the names in Figure 2 in Section 3.2.

The corrected section reads (lines 113-122): 

2.2. Chemicals

The main reagent standards (purity≥98%) employed in this analysis included theophylline, caffeine, gallic acid, (+)-catechin, (-)-catechin gallate (-)-epigallocatechin, (-)-epicatechin gallate, (-)-epigallocatechin, (-)-epigallocatechin gallate, (+)-gallocatechin, (-)-gallocatechin gallate, quercetin, rutin, kaempferol, myrcene, and ethyl decanoate, procured from Shanghai Yuanye Biotechnology Co. Anthrone (ACS reagent) and nin-hydrin (ACS reagent) were sourced from Sigma-Aldrich Company, USA. Methanol and acetonitrile (all chromatographic grade) were purchased from Thermo Fisher Company, USA. All remaining reagents were of analytical grade and obtained from Tianjin Damao Chemical Reagent Factory.

The corrected section 3.2 is from lines 258 to 266:

Figure 2. WE: Water extracts, FAA: Free amino acids, SS: Soluble sugars, TP: Tea polyphenol, TF: Theaflavins, TR: Thearubigins, TB: Theabrownins, CA: Caffeine, C: (+)-catechin, CG: (-)-catechin gallate, EC: (-)-epicatechin, ECG: (-)-epicatechin gallate, EGC: (-)-epigallocatechin, EGCG: (-)-epigallocatechin gallate, GC: (+)-gallocatechin, GCG: (-)-gallocatechin gallate, GA: Gallic acid, Theo: Theophylline, Que: Quercetin, Rut: Rutin, Kae: Kaempferol, Myr: Myrcene. (A) Heat dia-gram of the non-volatile compounds. (B) Scatter plot for the PCA model (total). (C) Wayne dia-grams of potential makers, filter conditions are VIP > 1, FC > 1.5 or < 0.8. Blue labelled compounds were highest in group A1 and purple labelled compounds were highest in group C1.

Comments 7: Line 243 is missing a space at the beginning of the sentence.

Reponse 7: A missing space at the beginning of line 243 has been added. The corrected sentence reads (line 255): …..from Group B1. These evolving characteristics likely contribute to the unique regional flavour and allow for differentiation based on storage duration of RPT (Table S1).

Comments 8: Why do they put the names of the compounds to be analysed in figure 3 and not in figure 2? The graphs in figure 3 look very blurry, perhaps they should be included as supplementary information.

Reponse 8: Dear reviewer, we have moved the names of the analyzed compounds from Figure 3 to Figure 2. Additionally, regarding the issue with Figure 3 being too large and requiring magnification to be clear, we have relocated Figure 3 to the supplementary materials. The figure is named as Figure S1.

Comments 9: I think there is a nomenclature error, in some parts A1, A2, A3 are mentioned and in others A1, B1 and C1. Improve quality of figure 5.

Reponse 9: We apologize for the mistake. We have corrected the incorrect labels in Figure 2 from A1, A2, A3 to the correct labels A1, B1, C1. We have also improved the quality of Figure 5.

Comments 10: Line 373 is missing space in the text. Review the entire document as this error occurs very frequently.

Reponse 10: A missing space in the text of line 373 has been added. The corrected sentence reads (lines 382-384): In addition, 2-Methyl-Naphthalene (OAV = 2729.03, sweet, floral, woody) and cis-Anethol (OAV = 578.85, herbal, aniseed, sweet) displayed higher OAVs in B2 com-pared to the tea samples from group C2 (Figure 6) (Table S4). Drawing on your valuable comment, we have checked the text formatting throughout the text. Moreover, the missing spaces throughout the text have been added.

Finally, we sincerely express our heartfelt gratitude for the review work you have done for us.

Reviewer 2 Report

Comments and Suggestions for Authors

Title of the reviewed manuscript: Characteristic Changes and Potential Markers of Flavour in Raw Pu-erh Tea with Different Ageing Cycles Analysed by HPLC, HS-SPME-GC-MS and OAV.

This study examined the changes in flavor characteristics and quality of raw Pu-erh tea throughout a 10-year storage period in Wenshan Prefecture, Yunnan Province, in order to elucidate the unique markers of flavor in different ageing cycles.

The research topic is interesting but the manuscript has several crucial flaws that need to be corrected considerably.

As the authors stated (line 501) and according the previous research (Xu et al., 2021) the aroma of raw Pu-erh tea is affected by the storage environment. There is a lack of information about storage conditions during 1-10 years of samples storage. Also, how RPTs were maintained before further processing?

In my opinion, information about instruments and equipment should be included in sections describing the methods in which they are applied, and section 2.3. should be omitted.

In the sensory evaluation section (lines 126-135), the sample size (5 experts) is relatively small. It is recommended to increase the number of assessors or provide reasonable evidence for the sample size.

Sensory evaluation is described very generally. Which method was used for sensory evaluation? Whether evaluators have received professional training on RPT evaluation? How the parameters were scored? I suggest analyzing the description and supplementing it.

lines 133-134: ...“This brewing process was repeated twice”... Please clarify this. In previous sentence is stated that 150 ml of boiling water was added in two batches.  

Figure 1 is hard to read. The resolution should be increased.

lines 137-139: ...“ Water extract (WE), tea polyphenols (TP), free amino acids (FAA), soluble sugars (SS), theaflavins (TF), thearubigins (TR), and theabrownins (TB) content were analysed following the spectrophotometric method described by Wang [20].”... Please rewrite the sentence. Only TF, TR and TR were analysed following the spectrophotometric method described by Wang et al. [20]

lines 142-143: ...” Soluble sugar content was determined utilising the anthrone-sulfuric acid colourimetric method [18]”... Xu et al. [18] analyzed volatile metabolites in raw Pu-erh tea and there is no description of mentioned method. Method should be adequately described.

lines 142-146: ...”Soluble sugar content was determined utilising the anthrone-sulfuric acid colourimetric method [18] and according to GB/T8305-143 2013, "Determination of Tea Water Leachate," which outlines the determination of water leachate, theaflavin, thearubigin, and theabrownin content through systematic analysis [18].”... The sentence is not clear. Please clarify it.

line 186: ...” among other tools.”... What other tools?

line 450: ...” FC > 1.8 or < 0.8”... It should be 1.5

The manuscript should be checked for typos, e.g. line 21: years(1-10); line 41: plant[1]; line 44: tea[2,3]; line 46: effects[4-7]; line86: et al; line 126: Evaluatio; line 458: aaroma; line 465: carotenoids.[41]. etc.

The form of presentation of the results, as well as the methodological description, in my opinion require refinement.

Comments on the Quality of English Language

I wanted to recommend that the manuscript's English writing should be improved for clarity and flow. A polished version would help make the text more understandable and improve its overall readability.

Author Response

Dear Reviewer:

We are very grateful for your praise of the subject in our article and for the professional review work you have done. As you are concerned, there are several problems that need to be addressed. According to your nice suggestions, we have made extensive corrections to our previous draft, the detailed corrections are listed below.

Comments 1: As the authors stated (line 501) and according the previous research (Xu et al., 2021) the aroma of raw Pu-erh tea is affected by the storage environment. There is a lack of information about storage conditions during 1-10 years of samples storage. Also, how RPTs were maintained before further processing?

Reponse 1: Dear Reviewer, following your suggestions, we have corrected the information on storage conditions and the maintenance of samples before further processing in lines 105-110. The corrected sentence reads (lines 106-111): The production followed the processing method stipulated in GB/T22111-2008, and the products were then uniformly stored in a dry and well-ventilated warehouse located in Wenshan Prefecture, with a relative humidity of ≤ 70% and an indoor temperature of ≤ 25 °C. Before further processing, each raw Pu-erh tea sample was ground and sieved through 60-mesh, tea samples was pulverized and kept at -20 °C.

Comments 2: In my opinion, information about instruments and equipment should be included in sections describing the methods in which they are applied, and section 2.3. should be omitted.

Reponse 2: Dear reviewer, according to your suggestion, we have omitted Section 2.3 and included the information about instruments and equipment in the sections describing their application methods. 

The specific corrections are made in lines 147-149 of Section 2.4.: The measurements using the above methods were all carried out on a UV-2102PC spectrophotometer (Element Analytical Instrument Co., Ltd., Shanghai, China).

Lines 152-154 of Section 2.4.:The instrument used in this study is a 1200 high-speed liquid chromatograph equipped with a C18 column (4.6 mm × 100 mm, 2.7 µm, Agilent, USA).

Lines 165-168 of Section 2.5.:The instruments used in this study include a 7890A-5975C headspace solid-phase mi-croextraction GC-MS (Agilent, USA), a DB-WAX column (30 m × 0.25 mm × 0.25 µm, Agilent, USA), and a 65 µm solid-phase microextraction head (PDMS/DVB, Supelco, USA).

Comments 3: In the sensory evaluation section (lines 126-135), the sample size (5 experts) is relatively small. It is recommended to increase the number of assessors or provide reasonable evidence for the sample size.

Reponse 3: Dear reviewer, thank you for your valuable suggestions. The experts involved in this sensory evaluation were five experienced members (three females and two males, aged between 22 and 55 years old) from the Tea College of Yunnan Agricultural University. They have accumulated 3–30 years of rich professional experience in tea sample sensory evaluation and have obtained the corresponding tea evaluation qualifications. Prior to conducting the sensory evaluation, we reviewed the relevant literature, where the number of expert members was also five [1, 2]. We sincerely appreciate your valuable suggestions.

  1. Chen, Z.; Dai W.; Xiong M.; Gao J.; Zhou H.; Chen D.; Li Y.Metabolomics investigation of the chemical variations in white teas with different producing areas and storage durations. Food Chem. X 2024, 21, 
  2. Deng, X.; Huang, G.; Tu, Q.; Zhou, H.; Li, Y.; Shi, H.; Wu, X.; Ren, H.; Huang, K.; He, X. Evolution analysis of flavour-active compounds during artificial fermentation of Pu-erh tea. Food Chem2021, 357, 129783.

Comments 4: Sensory evaluation is described very generally. Which method was used for sensory evaluation? Whether evaluators have received professional training on RPT evaluation? How the parameters were scored? I suggest analyzing the description and supplementing it.

Reponse 4: Dear reviewer, We conducted the sensory evaluation according to China's national standard GBT 23776-2018 ‘Methodology for Sensory Evaluation of Tea’. This method has been indicated in lines 123-125 of Section 2.3 Sensory Evaluation: The sensory evaluation of raw Pu-erh tea samples from different storage years in Wenshan Prefecture was conducted following China's national standard GBT23776-2018 "Methodology for Sensory Evaluation of Tea".

Before the evaluation, the evaluators had received professional training in RPT (assessment and were highly experienced. Meanwhile, we have added explanations in lines 128-130: Prior to the assessment, The group members received professional training in the evaluation of raw Pu-erh tea in accordance with the "Methodology for Sensory Evalua-tion of Tea".

The scoring method for the parameters has been added to lines 136-139: Ultimately, the group's assessment data were compiled; subsequently, the quality of the tea samples was quantitatively evaluated using the prescribed formula. The formula is as follows:

PRT (total score) = 20% × (a) + 30% × (b) + 10% × (c) +35% × (d) + 5% × (e)

    And We have added the quantitative scores of the sensory evaluation results to Table 2 in the manuscript (lines 214-216):

    Table 2. The sensory evaluation scoring outcomes of raw Pu-erh tea with different storage years in Wenshan Prefecture, Yunnan Province.

sample

appearance

s

aroma

s

soup color

s

taste

s

Tea residue

s

total

Y1

 Dark green

94

Faint scent with sweet, strong and lasting (+)

96

Yellowish-green, bright

92

Strong and mellow, astringent

88

Yellowish-green

92

92.20

Y2

Auburnish-green

94

Faint scent with sweet (+), strong

92

greenish yellow,

bright

92

Strong and mellow, little astringent

90

Greenish-yellow

92

91.70

Y3

Greenish-auburn

94

Sweet, less pure

78

Light yellow, bright (+)

94

Nearly strong and mellow, little astringent

86

Greenish-yellow (+)

94

86.40

Y4

Brownish-yellow

92

Woody, approach pure

80

Yellow, bright (+)

94

Approach mellow, little astringent

78

Auburnish-green

92

83.70

Y5

Yellowish-brown

92

Woody, pure,lasting

92

Yellow (+), bright (+)

94

Approach sweet and mellow, little astringent

78

Auburnish-green (+)

94

87.40

Y6

Auburnish-yellow

92

Ageing with sweet, strong and lasting

94

Deep yellow, bright

92

Nearly sweet and mellow, little astringent

86

Auburnish-yellow

92

90.50

Y7

Yellowish-auburn

92

Ageing with sweet, strong

92

Orange, bright

92

Mellow and sweet, little astringent

90

Auburnish-yellow (+)

94

91.40

Y8

Yellowish-auburn (+)

94

Ageing, more strong

90

Orange, bright

92

Mellow (+) and sweet

94

Yellowish-auburn

92

92.50

Y9

Auburnish-brown

92

Ageing with sweet, strong and lasting (+)

96

Orange, bright

92

Mellow (++) and sweet

95

Yellowish-auburn(+)

94

94.35

Y10

Brownish-auburn

92

Ageing with sweet, strong and lasting

94

Orange (+)

90

Mellow and thick and sweet

92

Yellowish-auburn (++)

95

92.55

Comments 5: lines 133-134: ...“This brewing process was repeated twice”... Please clarify this. In previous sentence is stated that 150 ml of boiling water was added in two batches.  

Reponse 5: Dear reviewer, thank you for pointing out the grammatical errors in our manuscript. After reviewing the research methods, we have corrected the content and expression of this error in lines 130-132: For each evaluation, 3 g of tea were placed in a professional evaluation cup, and added to 150 ml of boiling water was added twice, steeping for 2 minutes on the first and 5 minutes on the second.

Comments 6: Figure 1 is hard to read. The resolution should be increased.

Reponse 6: Dear reviewer, following your suggestion, we have made every effort to address the resolution issue of Figure 1. The problem has been somewhat improved after the adjustment. The text content can be seen more clearly when the figure is enlarged.

Comments 7: lines 137-139: ...“ Water extract (WE), tea polyphenols (TP), free amino acids (FAA), soluble sugars (SS), theaflavins (TF), thearubigins (TR), and theabrownins (TB) content were analysed following the spectrophotometric method described by Wang [20].”... Please rewrite the sentence. Only TF, TR and TR were analysed following the spectrophotometric method described by Wang et al. [20]

Reponse 7: Dear reviewer, upon reviewing this sentence, we also identified some unclear expressions. We have therefore removed this paragraph and rephrased the content more clearly and concisely in lines 141-147: Tea polyphenol content was determined according to GB/T8313-2018, "Determina-tion of Tea Polyphenols and Catechins in Tea". Free amino acids content was determined according to GB/T8314-2013, "Determination of Total Free Amino Acids in Tea". The content of water extracts was determined in accordance with GB/T 8305-2013, "Deter-mination of Water Extract in Tea". The content of soluble sugar was determined by the anthrone-sulfuric acid colorimetric method, and the contents of theaflavins, thearu-bigins, and theabrownins were measured by the system analysis method [20].

Comments 8: lines 142-143: ...” Soluble sugar content was determined utilising the anthrone-sulfuric acid colourimetric method [18]”... Xu et al. [18] analyzed volatile metabolites in raw Pu-erh tea and there is no description of mentioned method. Method should be adequately described.

Reponse 8: Dear reviewer, we sincerely apologize for this mistake. The method cited was actually from Wang’s study [20]. During the manuscript preparation, we carelessly retained the citation order from the initial draft. We have made the correction in lines 145-147: The content of soluble sugar was determined by the anthrone-sulfuric acid colorimetric method, and the contents of theaflavins, thearubigins, and theabrownins were meas-ured by the system analysis method [20].

Comments 9: lines 142-146: ...”Soluble sugar content was determined utilising the anthrone-sulfuric acid colourimetric method [18] and according to GB/T8305-143 2013, "Determination of Tea Water Leachate," which outlines the determination of water leachate, theaflavin, thearubigin, and theabrownin content through systematic analysis [18].”... The sentence is not clear. Please clarify it.

Reponse 9: Dear reviewer, we apologize for the unclear expression of this sentence. We have made corrections in lines 143-147: The content of water extracts was determined in accordance with GB/T 8305-2013, "Determination of Water Extract in Tea". The content of soluble sugar was determined by the anthrone-sulfuric acid colorimetric method, and the contents of theaflavins, thearubigins, and theabrownins were measured by the system analysis method [20].

Comments 10: line 186: ...” among other tools.”... What other tools?

Reponse 10: Dear reviewer, other tools used are Excel and PowerPoint, which are mainly used for the creation of wheel figure. We have corrected the sentence in lines 194-195: Data visualisation and representation were achieved with Origin 2021, MetaboAnalyst, TBtools, Chiplot. Online, Excel and PowerPoint.

Comments 11: line 450: ...” FC > 1.8 or < 0.8”... It should be 1.5

Reponse 11:Dear reviewer, the corrected reads ( line 461): ...VIP > 1, FC > 1.5 or < 0.8...

Comments 12: The manuscript should be checked for typos, e.g. line 21: years(1-10); line 41: plant[1]; line 44: tea[2,3]; line 46: effects[4-7]; line86: et al; line 126: Evaluatio; line 458: aaroma; line 465: carotenoids.[41]. etc.

Reponse 12: Dear reviewer, we sincerely apologize for this oversight and have already made the necessary corrections: Line 21: years (1-10 years); line 41: plant [1]; lines 44-45: Pu-erh tea [2,3]; line 46-47: effects [4-7]; line87: et al.; line 123: Evaluation; line 468: aroma; line 475-676: carotenoids [43].

Comments 13: The form of presentation of the results, as well as the methodological description, in my opinion require refinement.

Reponse 13: Dear reviewer, we have revised the form of presentation of the results according to the suggestions you mentioned above. Thank you sincerely for the review work you have done for us.

Comments 14: I wanted to recommend that the manuscript's English writing should be improved for clarity and flow. A polished version would help make the text more understandable and improve its overall readability.

Reponse 14: Dear reviewer, thank you for your suggestions. We have revised the English writing of the entire text according to your advice, including polishing sentences, using punctuation marks correctly, and correcting grammatical errors. TThe grammatical errors are marked in red in the text, for example:

line 14 Yunnan Organic lea corrected as: Yunnan Organic tea

line 26 Yunnan Organic leawhich also accompanied by that the colour changes from green to orange or brown, corrected as: .....which is also accompanied by that the colour changes from green to orange or brown,....

line 27 Ageing corrected as: ageing

line 62-64 At present, there have been many studies exploring the internal mechanisms of raw Pu-erh tea ageing across different storage durations, and analysing the reasons for the quality differences in raw Pu-erh tea. Corrected as: At present, numerous studies have explored the internal mechanisms of raw Pu-erh tea ageing across different storage years, and analysed the reasons for the qual-ity differences in raw Pu-erh tea.

line 126 'Methodology for sensory evaluation of tea' corrected as: "Methodology for Sensory Evaluation of Tea"

line 126 Five experienced experts,...corrected as: Five experienced members

line 130 For each evaluation, 3 g of tea was placed in corrected as :For each evaluation, 3 g of tea were placed in...

line 144 water extract corrected as: water extracts.

Thank you sincerely for the review work you have done for us.

Reviewer 3 Report

Comments and Suggestions for Authors

The authors submitted an original research paper of interest that dealt with the volatile and non-volatile constituents of raw Pu-erh tea, as well as the changes in these constituents during the storage of the tea. Given the extensive popularity of tea and its well-documented health benefits, the manuscript's subject matter is important within the field and relevant to current research trends. The results appear to support the constituents. The authors cited the most relevant refences.

The authors employed rigorous analytical methods, including gas and liquid chromatography, to comprehensively analyze the constituents of tea. Furthermore, the experimental data were evaluated by appropriate statistical methods, e.g., PCR. The results were presented with numerous illustrative figures. The results appear to support the constituents.

However, a minor issue has been identified: It is recommended that the resolution of Figure 3 be improved.

Author Response

Dear Reviewer:

We are very grateful for your praise of this topic in our article and your affirmation of the research content, as well as for the professional review work you have done. Following your suggestions, we have improved the resolution of Figure 3 in the previous manuscript. Due to the large file size, we have placed the adjusted figure in the supplementary materials, where it is named Figure S1. Thank you sincerely for the review work you have done for us.

Reviewer 4 Report

Comments and Suggestions for Authors

The manuscript "Characteristic Changes and Potential Markers of Flavour in Raw Pu-erh Tea with Different Ageing Cycles Analysed by HPLC, HS-SPME-GC-MS and OAV" focuses on monitoring the taste characteristics and quality changes of raw Pu-erh tea during storage for 1 to 10 years. It also constructs a sensor wheel of taste, providing theoretical support for the quality control of this raw material on the market and in warehouses.

Comment 1: Apart from the length of storage, what other factors can affect the changes in the sensory and qualitative characteristics of Raw Pu-erh tea? Please state them.

Comment 2: I don't understand the purpose of Figure 5. It's not mentioned in the text.

General comment: The innovativeness of the manuscript lies in the fact that it could serve as a benchmark for determining the market price of this tea.

Author Response

Dear Reviewer:

We are very grateful for your praise of this topic in our article and your affirmation of the research content, as well as for the professional review work you have done. As you are concerned, there are several problems that need to be addressed. According to your nice suggestions, we have made extensive corrections to our previous draft, the detailed corrections are listed below.

Comments 1: Apart from the length of storage, what other factors can affect the changes in the sensory and qualitative characteristics of Raw Pu-erh tea? Please state them.

Reponse 1: Dear reviewer, we have addressed this issue in lines 80-91 of Section 1 (Introduction): Ma [16] utilised multivariate statistical methods to analyse the aroma composition of raw Pu-erh tea from Lincang City, a clear distinction could be observed between sam-ples originating from the southern and northern regions of Lincang City. Southern samples were represented by relatively high concentrations of alcohols and esters, whereas northern samples displayed higher levels of aldehydes, ketones, and other substances, the altitude was hypothesised as a potentially significant factor contrib-uting to the observed regional variations in aroma profiles [17]. Xu et al. concluded that the aroma of raw Pu-erh tea is affected by the storage environment. In a dry and cold storage environment, raw Pu-erh tea is more likely to produce floral and sweet aromas. In contrast, in a hot and humid environment, raw Pu-erh tea is more likely to produce woody and aged aromas [18].

And lines 510-512 of Section 5 (Conclusions): Furthermore, RPT quality is affected by a variety of factors, including harvest season, raw material grade, processing methods, and storage conditions, future studies with larger sample sizes and enhanced variable control are warranted. Additionally, we have provided detailed supplements on the selection and storage conditions of the samples in lines 103-111 of Section 2.1 Tea Samples: Samples stored for 1-10 years were all offered by Longli Chun Tea Industry Co., Ltd., located in Wenshan Zhuang and Miao Autonomous Prefecture, Yunnan Province, in China. All samples were produced from Yunnan large-leaf varietals, utilising one bud and two to three leaves as fresh leaf raw material. The production followed the pro-cessing method stipulated in GB/T22111-2008, and the products were then uniformly stored in a dry and well-ventilated warehouse located in Wenshan Prefecture, with a relative humidity of ≤ 70% and an indoor temperature of ≤ 25 °C. Before further pro-cessing, each raw Pu-erh tea sample was ground and sieved through 60-mesh, tea samples was pulverized and kept at -20 °C.

Comments 2: I don't understand the purpose of Figure 5. It's not mentioned in the text.

Reponse 2: Dear reviewer, figure 5 is intended to help readers better understand the sentences in lines 300-301 and lines 303-304. Based on your comments, we have added clearer annotations to these two sentences.

lines 300-301: A total of 44 aroma-active compounds (OAV ≥ 1) were identified based on their odour thresholds and aroma profiles (Figure 4).

lines 303-304: The aroma characteristics of these 44 compounds fell into 9 descriptive categories: floral, fruity, woody, sweet, green, herbal, nuts, fatty and roasted (Figure 4).

Thank you sincerely for the review work you have done for us.

Round 2

Reviewer 1 Report

Comments and Suggestions for Authors

Based on the authors' comments, the modifications seem adequate to me, since the new document that was uploaded is a single-page document, I don't know if there is any error.

Author Response

Dear Reviewer,

When we revised the manuscript last time, we deleted the original content that was changed, and we apologize for any confusion this may have caused in your review. In this revision, we have retained the original content that was previously deleted. Meanwhile, since the line numbers have changed, we have copied the comments from the first round of review but updated the line numbers accordingly. Thank you once again for your review of our manuscript.

Comments 1: Line 47, there are double spaces before and after the text with “”.

Reponse 1: In line 47, the double space before and after “” has been deleted. The corrected sentence reads (line 51): At the same time, the characteristic of “the more it ages, the more fragrant it is” adds unique enjoyment to the Pu-erh tea and makes it a good collector's item [8].

Comments 2: I feel that the selection of samples with different aging ages is not entirely correct, because being the same plant, same cultivation area, same company, the climatic effects on the crops in different years cannot be controlled. More heat, more cold, more or less environmental humidity, same conditions or concentration of nutrients in the soil, pests, rain, droughts… all these are factors that alter the metabolism of the plant and consequently the accumulation of secondary metabolites, sugars, proteins among other elements of the leaf, and of course it can alter the final product, as well as its flavor, concentration of volatile metabolites and many others. It would have been interesting to analyze, for example, 3 drinks of 10 years, from different years, 3 drinks of 9 years from different years and so on. Something similar to wines, not every year the wines come out the same, despite being from the same vineyards, processes and applied technologies… there are always years that have better quality than others due to the effect of the climate.

Reponse 2: Dear Reviewer, thank you very much for your valuable suggestions on our study. Your insights regarding the impact of climatic conditions in different years on tea quality are very perceptive. We also agree with these insights. Indeed, climatic conditions (such as temperature, humidity, and rainfall) can significantly affect the metabolism of plants, thereby influencing the chemical composition and sensory quality of tea.

In our study, we chose samples from different years but from the same origin and company to minimize the variables caused by differences in cultivation conditions. This allows us to more accurately assess the effects of storage time on the flavour and quality of raw Pu-erh tea. Although we acknowledge that different climatic conditions in different years can have some impact on the initial quality of raw Pu-erh tea, the focus of our study is to explore the chemical and sensory changes that occur in tea during long-term storage under the same storage conditions. Regarding your suggestion of “comparing samples from different years,” we understand your point of view. In fact, we have also considered this in our study. The samples we selected all come from the standardized production batches of the same company over the years. Although it is not possible to completely eliminate the differences between years, this design helps us better understand the long-term impact of storage time on tea quality. In addition, we have reviewed a number of previous studies [1-6]. Excluding the influence of climatic conditions on tea metabolism, there are still certain patterns of metabolic changes in tea samples from different years. Therefore, we believe that the sample selection in this study is feasible.

We are very grateful for your comments. We have discussed the potential impact of climatic conditions on the initial quality of tea. Moreover, we believe that these suggestions will be of great reference value for our future experimental design and data analysis. In subsequent studies, we will consider using a more diverse range of samples, such as analyzing multiple batches of samples from different years, to more comprehensively assess the impact of climatic factors on tea quality.

We sincerely thank you for your valuable comments.

References:

  1. Guo, J.; Yu, Z.; Liu, M.; Guan, M.; Shi, A.; Hu, Y.; Li, S.; Yi, L.; Ren, D. Analysis of volatile profile and aromatic characteristics of raw Pu-erh tea during storage based on GC-MS and odour activity value.Foods 2023, 12, 3568.
  2. Zhou, B.; Ma, C.; Wu, T.; Xu, C.; Wang, J.; Xia, T. Classification of raw Pu-erh teas with different storage time based on characteristic compounds and effect of storage environment. Lwt2020, 133, 109914.
  3. Ma B.;Ma C.; Zhou B.; Chen X.; Wang Y.; Li Y.; Yin J.; Li X.; Quantitative descriptive analysis, non-targeted metabolomics and molecular docking reveal the dynamic aging and taste formation mechanism in raw Pu-erh tea during the storage. Food Chem. X, 2025,
  4. Characterization of volatile metabolites in Pu-erh teas with different storage years by combining GC-E-Nose, GC–MS, and GC-IMS.
  5. Chen Z.;Dai W.; **ong M.; Gao J.; Zhou H.; Chen D.; Li Y.; Metabolomics investigation of the chemical variations in white teas with different producing areas and storage durations. Food Chem. X, 2024, 21,
  6. Liang Y.; Liu Z.; Pang Y.; LiM.; Zheng S.; Pan F.; Guo C.; Wu Q.; Chen T.; Li Q.; Liu Z.; Effects of storage durations on flavour and bacterial communities in Liupao tea. Food Chem, 2025, 470,   

Comments 3: Line 128 “2.4. Sensory Evaluation” the letter “n” is missing in evaluation.

Reponse 3: Dear reviewer, a missing ‘n’ has been added to the title. The title has been corrected to reads ( line 153): 2.4. Sensory Evaluation.

Comments 4: It is not entirely clear whether aged tea is the plant alone, like the dried and aged plant, or whether it is aged prepared tea. Based on the sensory methodology, I think it is the aged plant alone. Please make it a little clearer.

Reponse 4: Dear reviewer, Here,the term "aged tea" refers to the storage duration of the tea samples, not to the plant alone. We apologize for the unclear explanation and thank you for pointing this out.

Comments 5: The image in figure 1 does not allow the text of the results to be read, and I expected the results to be quantitative (a scale of 1 to 5, for example) in order to differentiate the results, in terms of aroma, flavour, appearance...

Reponse 5: Dear reviewer, thank you for your suggestion. We have added the quantitative scores of the sensory evaluation results to Table 2 in the manuscript (lines 273-275):

Table 2. The sensory evaluation scoring outcomes of raw Pu-erh tea with different storage years in Wenshan Prefecture, Yunnan Province.

sample

appearance

s

aroma

s

soup color

s

taste

s

Tea residue

s

total

Y1

 Dark green

94

Faint scent with sweet, strong and lasting (+)

96

Yellowish-green, bright

92

Strong and mellow, astringent

88

Yellowish-green

92

92.20

Y2

Auburnish-green

94

Faint scent with sweet (+), strong

92

greenish yellow,

bright

92

Strong and mellow, little astringent

90

Greenish-yellow

92

91.70

Y3

Greenish-auburn

94

Sweet, less pure

78

Light yellow, bright (+)

94

Nearly strong and mellow, little astringent

86

Greenish-yellow (+)

94

86.40

Y4

Brownish-yellow

92

Woody, approach pure

80

Yellow, bright (+)

94

Approach mellow, little astringent

78

Auburnish-green

92

83.70

Y5

Yellowish-brown

92

Woody, pure,lasting

92

Yellow (+), bright (+)

94

Approach sweet and mellow, little astringent

78

Auburnish-green (+)

94

87.40

Y6

Auburnish-yellow

92

Ageing with sweet, strong and lasting

94

Deep yellow, bright

92

Nearly sweet and mellow, little astringent

86

Auburnish-yellow

92

90.50

Y7

Yellowish-auburn

92

Ageing with sweet, strong

92

Orange, bright

92

Mellow and sweet, little astringent

90

Auburnish-yellow (+)

94

91.40

Y8

Yellowish-auburn (+)

94

Ageing, more strong

90

Orange, bright

92

Mellow (+) and sweet

94

Yellowish-auburn

92

92.50

Y9

Auburnish-brown

92

Ageing with sweet, strong and lasting (+)

96

Orange, bright

92

Mellow (++) and sweet

95

Yellowish-auburn(+)

94

94.35

Y10

Brownish-auburn

92

Ageing with sweet, strong and lasting

94

Orange (+)

90

Mellow and thick and sweet

92

Yellowish-auburn (++)

95

92.55

Comments 6: From my perspective as a reader, I feel that the nomenclature of the bioactive compounds described in point 2.2 does not help in reading the results because there are many nomenclatures, some very similar and it is difficult to compare the variations using the nomenclature, it would be ideal to have the names, for example, in section 3.2.

Reponse 6: Thank you for your valuable comments. To improve the readability of the results, we have removed the nomenclature in Section 2.2 and adopted a more comprehensive description of the compounds. Additionally, we have annotated the names in Figure 2 in Section 3.2.

The corrected section reads (lines 144-152): 

2.2. Chemicals

The main reagent standards (purity≥98%) employed in this analysis included theophylline, caffeine, gallic acid, (+)-catechin, (-)-catechin gallate (-)-epigallocatechin, (-)-epicatechin gallate, (-)-epigallocatechin, (-)-epigallocatechin gallate, (+)-gallocatechin, (-)-gallocatechin gallate, quercetin, rutin, kaempferol, myrcene, and ethyl decanoate, procured from Shanghai Yuanye Biotechnology Co. Anthrone (ACS reagent) and nin-hydrin (ACS reagent) were sourced from Sigma-Aldrich Company, USA. Methanol and acetonitrile (all chromatographic grade) were purchased from Thermo Fisher Company, USA. All remaining reagents were of analytical grade and obtained from Tianjin Damao Chemical Reagent Factory.

The corrected section 3.2 is from lines 258 to 266:

Figure 2. WE: Water extracts, FAA: Free amino acids, SS: Soluble sugars, TP: Tea polyphenol, TF: Theaflavins, TR: Thearubigins, TB: Theabrownins, CA: Caffeine, C: (+)-catechin, CG: (-)-catechin gallate, EC: (-)-epicatechin, ECG: (-)-epicatechin gallate, EGC: (-)-epigallocatechin, EGCG: (-)-epigallocatechin gallate, GC: (+)-gallocatechin, GCG: (-)-gallocatechin gallate, GA: Gallic acid, Theo: Theophylline, Que: Quercetin, Rut: Rutin, Kae: Kaempferol, Myr: Myrcene. (A) Heat dia-gram of the non-volatile compounds. (B) Scatter plot for the PCA model (total). (C) Wayne dia-grams of potential makers, filter conditions are VIP > 1, FC > 1.5 or < 0.8. Blue labelled compounds were highest in group A1 and purple labelled compounds were highest in group C1.

Comments 7: Line 243 is missing a space at the beginning of the sentence.

Reponse 7: A missing space at the beginning of line 243 has been added. The corrected sentence reads (line 318): …..from Group B1. These evolving characteristics likely contribute to the unique regional flavour and allow for differentiation based on storage duration of RPT (Table S1).

Comments 8: Why do they put the names of the compounds to be analysed in figure 3 and not in figure 2? The graphs in figure 3 look very blurry, perhaps they should be included as supplementary information.

Reponse 8: Dear reviewer, we have moved the names of the analyzed compounds from Figure 3 to Figure 2. Additionally, regarding the issue with Figure 3 being too large and requiring magnification to be clear, we have relocated Figure 3 to the supplementary materials. The figure is named as Figure S1.

Comments 9: I think there is a nomenclature error, in some parts A1, A2, A3 are mentioned and in others A1, B1 and C1. Improve quality of figure 5.

Reponse 9: We apologize for the mistake. We have corrected the incorrect labels in Figure 2 from A1, A2, A3 to the correct labels A1, B1, C1. We have also improved the quality of Figure 5.

Comments 10: Line 373 is missing space in the text. Review the entire document as this error occurs very frequently.

Reponse 10: A missing space in the text of line 373 has been added. The corrected sentence reads (lines 454-457): In addition, 2-Methyl-Naphthalene (OAV = 2729.03, sweet, floral, woody) and cis-Anethol (OAV = 578.85, herbal, aniseed, sweet) displayed higher OAVs in B2 com-pared to the tea samples from group C2 (Figure 6) (Table S4). Drawing on your valuable comment, we have checked the text formatting throughout the text. Moreover, the missing spaces throughout the text have been added.

Finally, we sincerely express our heartfelt gratitude for the review work you have done for us.

Reviewer 2 Report

Comments and Suggestions for Authors

Dear Authors,

It is obvious that my comments have been studied carefully and that the authors have acknowledged my suggestions and adequately responded to my comments.

However, uploaded revised version of manuscript available to me for review is not appropriate.

Author Response

Dear Reviewer,

When we revised the manuscript last time, we deleted the original content that was changed, and we apologize for any confusion this may have caused in your review. In this revision, we have retained the original content that was previously deleted. Meanwhile, since the line numbers have changed, we have copied the comments from the first round of review but updated the line numbers accordingly. Thank you once again for your review of our manuscript.

Comments 1: As the authors stated (line 501) and according the previous research (Xu et al., 2021) the aroma of raw Pu-erh tea is affected by the storage environment. There is a lack of information about storage conditions during 1-10 years of samples storage. Also, how RPTs were maintained before further processing?

Reponse 1: Dear Reviewer, following your suggestions, we have corrected the information on storage conditions and the maintenance of samples before further processing in lines 105-110. The corrected sentence reads (lines 125-130): The production followed the processing method stipulated in GB/T22111-2008, and the products were then uniformly stored in a dry and well-ventilated warehouse located in Wenshan Prefecture, with a relative humidity of ≤ 70% and an indoor temperature of ≤ 25 °C. Before further processing, each raw Pu-erh tea sample was ground and sieved through 60-mesh, tea samples was pulverized and kept at -20 °C.

Comments 2: In my opinion, information about instruments and equipment should be included in sections describing the methods in which they are applied, and section 2.3. should be omitted.

Reponse 2: Dear reviewer, according to your suggestion, we have omitted Section 2.3 and included the information about instruments and equipment in the sections describing their application methods. 

The specific corrections are made in lines 196-198 of Section 2.4.: The measurements using the above methods were all carried out on a UV-2102PC spectrophotometer (Element Analytical Instrument Co., Ltd., Shanghai, China).

Lines 201-202 of Section 2.4.:The instrument used in this study is a 1200 high-speed liquid chromatograph equipped with a C18 column (4.6 mm × 100 mm, 2.7 µm, Agilent, USA).

Lines 215-217 of Section 2.5.:The instruments used in this study include a 7890A-5975C headspace solid-phase mi-croextraction GC-MS (Agilent, USA), a DB-WAX column (30 m × 0.25 mm × 0.25 µm, Agilent, USA), and a 65 µm solid-phase microextraction head (PDMS/DVB, Supelco, USA).

Comments 3: In the sensory evaluation section (lines 126-135), the sample size (5 experts) is relatively small. It is recommended to increase the number of assessors or provide reasonable evidence for the sample size.

Reponse 3: Dear reviewer, thank you for your valuable suggestions. The experts involved in this sensory evaluation were five experienced members (three females and two males, aged between 22 and 55 years old) from the Tea College of Yunnan Agricultural University. They have accumulated 3–30 years of rich professional experience in tea sample sensory evaluation and have obtained the corresponding tea evaluation qualifications. Prior to conducting the sensory evaluation, we reviewed the relevant literature, where the number of expert members was also five [1, 2]. We sincerely appreciate your valuable suggestions.

  1. Chen, Z.; Dai W.; Xiong M.; Gao J.; Zhou H.; Chen D.; Li Y.Metabolomics investigation of the chemical variations in white teas with different producing areas and storage durations. Food Chem. X 2024, 21, 
  2. Deng, X.; Huang, G.; Tu, Q.; Zhou, H.; Li, Y.; Shi, H.; Wu, X.; Ren, H.; Huang, K.; He, X. Evolution analysis of flavour-active compounds during artificial fermentation of Pu-erh tea. Food Chem2021, 357, 129783.

Comments 4: Sensory evaluation is described very generally. Which method was used for sensory evaluation? Whether evaluators have received professional training on RPT evaluation? How the parameters were scored? I suggest analyzing the description and supplementing it.

Reponse 4: Dear reviewer, We conducted the sensory evaluation according to China's national standard GBT 23776-2018 ‘Methodology for Sensory Evaluation of Tea’. This method has been indicated in lines 163-165 of Section 2.3 Sensory Evaluation: The sensory evaluation of raw Pu-erh tea samples from different storage years in Wenshan Prefecture was conducted following China's national standard GBT23776-2018 "Methodology for Sensory Evaluation of Tea". 

Before the evaluation, the evaluators had received professional training in RPT (assessment and were highly experienced. Meanwhile, we have added explanations in lines 167-169: Prior to the assessment, The group members received professional training in the evaluation of raw Pu-erh tea in accordance with the "Methodology for Sensory Evalua-tion of Tea".

The scoring method for the parameters has been added to lines 175-178: Ultimately, the group's assessment data were compiled; subsequently, the quality of the tea samples was quantitatively evaluated using the prescribed formula. The formula is as follows:

PRT (total score) = 20% × (a) + 30% × (b) + 10% × (c) +35% × (d) + 5% × (e)

    And We have added the quantitative scores of the sensory evaluation results to Table 2 in the manuscript (lines 273-275):

    Table 2. The sensory evaluation scoring outcomes of raw Pu-erh tea with different storage years in Wenshan Prefecture, Yunnan Province.

sample

appearance

s

aroma

s

soup color

s

taste

s

Tea residue

s

total

Y1

 Dark green

94

Faint scent with sweet, strong and lasting (+)

96

Yellowish-green, bright

92

Strong and mellow, astringent

88

Yellowish-green

92

92.20

Y2

Auburnish-green

94

Faint scent with sweet (+), strong

92

greenish yellow,

bright

92

Strong and mellow, little astringent

90

Greenish-yellow

92

91.70

Y3

Greenish-auburn

94

Sweet, less pure

78

Light yellow, bright (+)

94

Nearly strong and mellow, little astringent

86

Greenish-yellow (+)

94

86.40

Y4

Brownish-yellow

92

Woody, approach pure

80

Yellow, bright (+)

94

Approach mellow, little astringent

78

Auburnish-green

92

83.70

Y5

Yellowish-brown

92

Woody, pure,lasting

92

Yellow (+), bright (+)

94

Approach sweet and mellow, little astringent

78

Auburnish-green (+)

94

87.40

Y6

Auburnish-yellow

92

Ageing with sweet, strong and lasting

94

Deep yellow, bright

92

Nearly sweet and mellow, little astringent

86

Auburnish-yellow

92

90.50

Y7

Yellowish-auburn

92

Ageing with sweet, strong

92

Orange, bright

92

Mellow and sweet, little astringent

90

Auburnish-yellow (+)

94

91.40

Y8

Yellowish-auburn (+)

94

Ageing, more strong

90

Orange, bright

92

Mellow (+) and sweet

94

Yellowish-auburn

92

92.50

Y9

Auburnish-brown

92

Ageing with sweet, strong and lasting (+)

96

Orange, bright

92

Mellow (++) and sweet

95

Yellowish-auburn(+)

94

94.35

Y10

Brownish-auburn

92

Ageing with sweet, strong and lasting

94

Orange (+)

90

Mellow and thick and sweet

92

Yellowish-auburn (++)

95

92.55

Comments 5: lines 133-134: ...“This brewing process was repeated twice”... Please clarify this. In previous sentence is stated that 150 ml of boiling water was added in two batches.  

Reponse 5: Dear reviewer, thank you for pointing out the grammatical errors in our manuscript. After reviewing the research methods, we have corrected the content and expression of this error in lines 169-171: For each evaluation, 3 g of tea were placed in a professional evaluation cup, and added to 150 ml of boiling water was added twice, steeping for 2 minutes on the first and 5 minutes on the second.

Comments 6: Figure 1 is hard to read. The resolution should be increased.

Reponse 6: Dear reviewer, following your suggestion, we have made every effort to address the resolution issue of Figure 1. The problem has been somewhat improved after the adjustment. The text content can be seen more clearly when the figure is enlarged.

Comments 7: lines 137-139: ...“ Water extract (WE), tea polyphenols (TP), free amino acids (FAA), soluble sugars (SS), theaflavins (TF), thearubigins (TR), and theabrownins (TB) content were analysed following the spectrophotometric method described by Wang [20].”... Please rewrite the sentence. Only TF, TR and TR were analysed following the spectrophotometric method described by Wang et al. [20]

Reponse 7: Dear reviewer, upon reviewing this sentence, we also identified some unclear expressions. We have therefore removed this paragraph and rephrased the content more clearly and concisely in lines 190-196: Tea polyphenol content was determined according to GB/T8313-2018, "Determina-tion of Tea Polyphenols and Catechins in Tea". Free amino acids content was determined according to GB/T8314-2013, "Determination of Total Free Amino Acids in Tea". The content of water extracts was determined in accordance with GB/T 8305-2013, "Deter-mination of Water Extract in Tea". The content of soluble sugar was determined by the anthrone-sulfuric acid colorimetric method, and the contents of theaflavins, thearu-bigins, and theabrownins were measured by the system analysis method [20].

Comments 8: lines 142-143: ...” Soluble sugar content was determined utilising the anthrone-sulfuric acid colourimetric method [18]”... Xu et al. [18] analyzed volatile metabolites in raw Pu-erh tea and there is no description of mentioned method. Method should be adequately described.

Reponse 8: Dear reviewer, we sincerely apologize for this mistake. The method cited was actually from Wang’s study [20]. During the manuscript preparation, we carelessly retained the citation order from the initial draft. We have made the correction in lines 194-196: The content of soluble sugar was determined by the anthrone-sulfuric acid colorimetric method, and the contents of theaflavins, thearubigins, and theabrownins were meas-ured by the system analysis method [20].

Comments 9: lines 142-146: ...”Soluble sugar content was determined utilising the anthrone-sulfuric acid colourimetric method [18] and according to GB/T8305-143 2013, "Determination of Tea Water Leachate," which outlines the determination of water leachate, theaflavin, thearubigin, and theabrownin content through systematic analysis [18].”... The sentence is not clear. Please clarify it.

Reponse 9: Dear reviewer, we apologize for the unclear expression of this sentence. We have made corrections in lines 192-196: The content of water extracts was determined in accordance with GB/T 8305-2013, "Determination of Water Extract in Tea". The content of soluble sugar was determined by the anthrone-sulfuric acid colorimetric method, and the contents of theaflavins, thearubigins, and theabrownins were measured by the system analysis method [20].

Comments 10: line 186: ...” among other tools.”... What other tools?

Reponse 10: Dear reviewer, other tools used are Excel and PowerPoint, which are mainly used for the creation of wheel figure. We have corrected the sentence in line 251: Data visualisation and representation were achieved with Origin 2021, MetaboAnalyst, TBtools, Chiplot. Online, Excel and PowerPoint.

Comments 11: line 450: ...” FC > 1.8 or < 0.8”... It should be 1.5

Reponse 11:Dear reviewer, the corrected reads ( line 541): ...VIP > 1, FC > 1.5 or < 0.8...

Comments 12: The manuscript should be checked for typos, e.g. line 21: years(1-10); line 41: plant[1]; line 44: tea[2,3]; line 46: effects[4-7]; line86: et al; line 126: Evaluatio; line 458: aaroma; line 465: carotenoids.[41]. etc.

Reponse 12: Dear reviewer, we sincerely apologize for this oversight and have already made the necessary corrections: Line 20: years (1-10 years); line 43: plant [1]; lines 48: Pu-erh tea [2, 3]; line 50: effects [4-7]; line 99: et al.; line 153: Evaluation; line 552: aroma; line 560: carotenoids [43].

Comments 13: The form of presentation of the results, as well as the methodological description, in my opinion require refinement.

Reponse 13: Dear reviewer, we have revised the form of presentation of the results according to the suggestions you mentioned above. Thank you sincerely for the review work you have done for us.

Comments 14: I wanted to recommend that the manuscript's English writing should be improved for clarity and flow. A polished version would help make the text more understandable and improve its overall readability.

Reponse 14: Dear reviewer, thank you for your suggestions. We have revised the English writing of the entire text according to your advice, including polishing sentences, using punctuation marks correctly, and correcting grammatical errors. TThe grammatical errors are marked in red in the text, for example:

line 9 M,Z. corrected as: M.Z.

line 13 Yunnan Organic lea corrected as: Yunnan Organic tea

line 25 which also accompanied by that the colour changes from green to orange or brown, corrected as: .....which is also accompanied by that the colour changes from green to orange or brown,....

line 26 Ageing corrected as: ageing

line 73-75 At present, there have been many studies exploring the internal mechanisms of raw Pu-erh tea ageing across different storage durations, and analysing the reasons for the quality differences in raw Pu-erh tea. Corrected as: At present, numerous studies have explored the internal mechanisms of raw Pu-erh tea ageing across different storage years, and analysed the reasons for the qual-ity differences in raw Pu-erh tea.

line 165 'Methodology for sensory evaluation of tea' corrected as: "Methodology for Sensory Evaluation of Tea"

line 165 Five experienced experts,...corrected as: Five experienced members

line 169 For each evaluation, 3 g of tea was placed in corrected as : For each evaluation, 3 g of tea were placed in...

line 193 water extract corrected as: water extracts.

Thank you sincerely for the review work you have done for us.